



**Measurement and model analyses of the ozone variation during 2006 to 2015 and its response**
**to emission change in megacity Shanghai, China**
Jianming Xu[1,2], Xuexi Tie[3,4], Wei Gao[1,2], Yanfen Lin[5], and Qingyan Fu[5]
[1] Yangtze River Delta Center for Environmental Meteorology Prediction and Warning, Shanghai
Meteorological Service, Shanghai, 200135, China
[2] Shanghai Key Laboratory of Health and Meteorology, Shanghai Meteorological Service, Shanghai,
200135, China
[3] Key Laboratory of Aerosol Chemistry & Physics, SKLLQG, Institute of Earth Environment, Chinese
Academy of Science, Xi'an, 710061, China
[4] Center for Excellence in Urban Atmospheric Environment, Institute of Urban Environment,
Chinese Academy of Science, Xiamen, 361021, China
[5] Shanghai Environmental Monitoring Center, Shanghai, 200135, China
Correspondence: Xuexi Tie (tiexx@ieecas.cn)

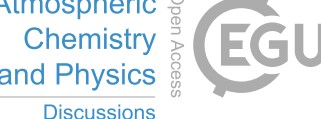



**Abstract.** The fine particles (PM$_{2.5}$) in China decrease significantly in recent years as a result
of the implement of Chinese Clean Air Action Plan since 2013, while the O$_3$ pollution is getting
worse, especially in megacities such as Beijing and Shanghai. Better understanding the elevated
O$_3$ pollution in Chinese megacities and its response to emission change is important for
developing an effective emission control strategy in future. In this study, we analyze the
significant increasing trend of O$_3$ concentration from 2006 to 2015 in the megacity Shanghai with
the variability of 1-1.3 ppbv yr$^{-1}$. It is likely attributed to the notable reduction of NO$_x$
concentration with the decreasing rate of 1.86-2.15 ppbv yr$^{-1}$ accompanied with the little change
of VOCs during the same period excluding the weak trends of meteorological impacts on local
dispersion (wind speed), regional transport (wind direction) and O$_3$ photolysis (solar radiation). It
is further illustrated by using a state of the art regional chemical/dynamical model (WRF-Chem)
to explore the O$_3$ variation response to the reduction of NO$_x$ emission in Shanghai. The control
experiment conducted in September of 2009 shows very excellent performance for O$_3$ and NO$_x$
simulations including both the spatial distribution pattern, and the day by day variation by
comparing with 6 in-situ measurements from MIRAGE-shanghai field campaign. Sensitive
experiments with 30% reduction of NO$_x$ emission from 2009 to 2015 in Shanghai estimated by
Shanghai Environmental Monitoring Center shows that the calculated O$_3$ concentrations exhibit
obvious enhancement by 4-7 ppbv in urban zones with the increasing variability of 0.96-1.06
ppbv yr$^{-1}$, which is well consistent with the observed O$_3$ trend as a result of the strong
VOC-limited condition for O$_3$ production. The large reduction of NO$_x$ combined with less change
of VOCs during the past ten years promotes the O$_3$ production in Shanghai to move towards
NO$_x$-limited regime. Further analysis of WRF-Chem experiments and O$_3$ isopleths diagram
suggests that the O$_3$ production in downtown is still under VOC-limited regime after 2015 despite
of the remarkable NO$_x$ reduction, while moves to the transition regime between NO$_x$-limited and
VOC-limited in sub-urban zones. Supposing the insignificant VOCs variation persists, the O$_3$
concentration in downtown would keep increasing till 2020 with the further 20% reduction of
NO$_x$ emission after 2015 estimated by Shanghai Clean Air Action Plan. While there are less O$_3$
change in other regions where the O$_3$ production is not under VOC-limited regime. The O$_3$
production in Shanghai will switch from VOC-limited to NO$_x$-limited regime after 2020 except
downtown area which is likely close to the transition regime. As a result the O$_3$ concentration will
decrease by 2-3 ppbv in sub-urban zones, and more than 4 ppbv in suburb response to 20%
reduction of NO$_x$ emission after 2020, whereas is not sensitive to both NO$_x$ and VOCs changes in
downtown. This result reveals that the control strategy of O$_3$ pollution is a very complex process,
and needs to be carefully studied.
**Key Words: O$_3$ pollution in Shanghai, Long-term O3 trend, WRF-Chem**

## 1 Introduction

Ozone ($O_3$) in the troposphere plays the important role in the oxidation of chemically and climatically relevant trace gases, hence regulating their lifetime in the atmosphere (Monks et al., 2015). In the lower troposphere, $O_3$ is produced from photochemical reactions involving volatile organic compounds (VOCs, broadly including CO) and nitrogen oxides ($NO_x$ = NO + $NO_2$) in the presence of sunlight (Brasseur et al., 1999). As a strong oxidant, $O_3$ at ground level is detrimental to human health and vegetation (Tai et al., 2014), and has been received continuous attention from both the scientific and regulatory communities in the past three decades.

Shanghai has emerged as one of the largest megacities in the world over the last two decades. The city has a fleet of over 3.6 million vehicles and the population of over 2400 million permanent residents, which results in high emissions of $NO_x$, VOCs, and primary particulate matter (PM) to the atmosphere from industrial and commercial activities, leading to the photochemical smog formation. Persistent high level of surface $O_3$ and PM were observed in Shanghai during the past ten years (Geng et al., 2007; Ran et al., 2009; Tie et al., 2009a; Xu et al., 2015). In order to mitigate the adverse impacts from severe air pollution, the Clean Air Action Plan was issued in the end of 2013 to implement the emission reduction program in Shanghai and its neighboring area. As a result, the annual mean $PM_{2.5}$ (particles with diameter $\leqq$ 2.5 μm) mass concentration has decreased from 50 μg m$^{-3}$ in 2013 to 39 μg m$^{-3}$ in 2017. However $O_3$ pollution has been continuously worsen, with the non-attainment days increased from 99 d in 2014 to 129 d in 2016. As a result, $O_3$ becomes the primary air pollutant affecting the ambient air quality instead of $PM_{2.5}$ in Shanghai. Similar issue has also been occurred in other cities in the eastern China (Lu et al., 2018). For example, the mean $PM_{2.5}$ mass concentration over the 74 major cites decreased by 40% from 2013 to 2017, whereas the maximum daily 8-h average $O_3$ concentration in summer exceeds the Chinese National Ambient Air Quality Stand over most of eastern China (Li et al., 2019). Thus better understanding the causes of elevated $O_3$ in China is important for developing effective $O_3$ control strategies, especially in megacities such as Shanghai.

A prerequisite to an effective emission-based $O_3$ control strategy is to understand the temporal and spatial relationship between $O_3$ and its precursors, and the response of $O_3$ concentrations to the changes in emissions of $O_3$-precursors (such as $NO_x$ and VOCs, Lin et al., 1988). The relationship of $O_3$ and $O_3$-precursors can be clarified as $NO_x$-limited or VOC-limited chemistry of $O_3$ formation, which is usually defined based on the relative impact of a given percent reduction in $NO_x$ relative to VOCs in the context of urban chemistry (Sillman, 1999).

Some observational and modeling works on $O_3$ chemical formation and transformation have been carried out in Shanghai since 2007. The $O_3$ production in Shanghai city is clearly under VOC-limited regime (Geng et al., 2007), in which the aromatics and alkenes play the dominant roles (Geng et al., 2008a). The aircraft measurements in Yangtze River Delta (YRD) region show the strong anti-correlation between $NO_x$ and $O_3$, indicating the similar VOC-limited regime for $O_3$ production in the area neighboring Shanghai (Geng et al., 2008b). Thus either $NO_x$ reduction or VOCs growth is favorable for $O_3$ enhancement in Shanghai. Gao et al. (2017) reported that $O_3$ concentration in Shanghai downtown increased 67% from 2006 to 2015, whereas $NO_x$ concentration decreased about 38%. This is also consistent with the results of Lin et al. (2017) that, the median of the maximum daily 8-h average $O_3$ concentration in Shanghai increased notably from 2006 to 2016, with the rate of 1.4 ppbv yr$^{-1}$, while the $NO_2$ decreased from 66.7 to



42.1 $\mu$g m$^{-3}$ with about 20% reduction. These previous studies provide the useful information
regarding the O$_3$ chemical formation and transformation in Shanghai. However, such O$_3$ variation
responding to emission change has not been clearly investigated. Considering that O$_3$ formation
is a complicated process including chemistry, transport, emission, deposition and their
interactions, the chemical transport model is the powerful tool to gain an understanding of these
interacting processes. For example, Lei et al. (2007), Ying et al. (2009) and Song et al. (2010)
investigated the O$_3$ production rate and its sensitivity to emission changes of O$_3$ precursors by
CAMx model in Mexico City Metropolitan Area (MCMA). Tie et al. (2013) analyzed the
comprehensive data of the MIRAGE-Shanghai field campaign by WRF-Chem model, and
quantified the threshold value by the emission ratio of NO$_x$/VOCs for switching from VOC-limited
to NO$_x$-limited in Shanghai. Recently Li et al. (2019) suggested an important cause of the
increasing O$_3$ in North China Plain (NCP) during 2013 to 2017 as the significant decrease in PM$_{2.5}$
slowing down the sink of hydroperoxy radicals and thus speeding up the O$_3$ production by
GOES-CHEM model. However such implication for O$_3$ trend is not pervasive in YRD and other
regions. Moreover, the 5-year O$_3$ records seem rather short to examine the inter-annual
variability of O$_3$ concentration. The GOES-CHEM experiment with 50 km resolution maybe is
suitable for the O$_3$ simulation at regional scale but is too coarse to resolve the local O$_3$ budget at
urban scale, such as Beijing or Shanghai. To our knowledge, there are no peer-reviewed modeling
studies focus on the past long term O$_3$ variations response to emission changes conducted in
Shanghai. Thus this paper extends the study of Tie et al. (2013) and Gao et al. (2017) to further
examine the inter-annual O$_3$ variations from 2005 to 2016 by long term measurements of O$_3$ and
its precursors in Shanghai. The effects of emission changes on long term O$_3$ variability are
evaluated by WRF-Chem model with high resolution and compared with measurements. The shift
of O$_3$ photochemical regime relative to the variations of NO$_x$ and VOCs concentrations in the past
ten years is discussed by O$_3$ isopleths diagram combined with WRF-Chem model to provide more
insights into the O$_3$ control strategy. Moreover, the future O$_3$ levels and its possible chemical
regime in Shanghai are also discussed according to the Shanghai Clean Air Action Plan.
The paper is constructed as follows. The measurements and models used for this study are
described in Sect. 2. The analysis on the long-term in-situ measurements of O$_3$ and its precursors,
as well as the model sensitive experiments are presented and discussed in Sect. 3-6. The
conclusion is summarized in Sect. 7.
**2 Measurements and models**
**2.1 Measurements**
The measurements of O$_3$ and NO$_x$ are collected from 6 sites (XJH, PD, JS, BS, SS, DT) over
Shanghai under different influence of air pollutant emissions. The XJH site is located at the
downtown of Shanghai, which is strongly influenced by emission of transportation. The PD site is
located at the sub-urban area near a big park, which is influenced by the mixed emissions of
transportation and residential. The JS site is located in the south of Shanghai with several large
chemical industries. The BS site is located in the north of Shanghai with some big steel and power
plants. The SS site is located at the top of the sole hill (100 m a.g.l) in Shanghai, which has minor
effect from regional emissions, and is influenced by regional transport. The DT site is located at a
remote island without anthropogenic activities. These O$_3$ and NO$_x$ measurements are used for



the evaluation on WRF-Chem performance. In addition, the VOCs are sampled at the downtown
site XJH and the sub-urban site PD, and are analyzed at a chemistry laboratory. The study on the
$O_3$ chemical production in this paper is limited at XJH and PD by the intensive measurements of
$O_3$ and its precursors (VOCs and $NO_x$) from 2006 to 2015 at these two urban sites. The
meteorological measurements including wind speed and direction, solar radiation and
temperature are collected at BS site, which is the only climatology observatory in Shanghai. The
meteorological measurements in BS are used for international exchange of meteorological data
representing Shanghai, sponsored by the World Meteorological Organization (WMO). The
geographical distribution of the 6 sites is indicated in Fig. 1.
**Figure 1.** (a) The distribution of topography height in Shanghai and its neighboring area. (b) The
distribution of land-use category in Shanghai. The locations of the 6 sites (XJH, BS, PD, SS, JS, DT)
are described by blue dots.

**2.2 Instruments**
$O_3$ is measured using an EC 9810 Ozone Analyzer, together with a UV photometer, which
accurately and reliably measures $O_3$ concentration in ambient air. The oxides of nitrogen analyzer
(EC9841B/ECOTECH) have a heated molybdenum $NO_2$ to NO converter. The resulting NO
concentration is quantified using the chemiluminescence technique. This instrument has
automated to set to be zero, and include an optional external valve manifold and external
calibration sources. Quality control checks are performed every 3 days, including inspection of
the shelter and instruments as well as zero, precision and span checks. Filter is replaced once
every two weeks and calibration is made every month. The $O_3$ concentrations are recorded every
1 min.
VOCs concentrations are sampled for 24 h every day with a 6 L silonite canister with a
silonite coated valve (model 29-10622). The internal silonite coating improves long-term VOC
storage. The instrument has a large volume to detect volatile chemicals down to low pptv range.
Absorption is eliminated by using nupropackless valves and by eliminating teflon tape in the valve
stem. These canisters are recognized to meet or exceed the technical specifications required for
EP methods TO14-A and TO15. Gases samples are pre-processed using Model 7100 VOC
preconcentrator. Samples are analyzed for VOCs using a gas chromatography system (Agilent
GC6890) coupled with mass-selective detection (Agilent MSD5975 N) with length of 60 m,
diameter of 0.32 mm, and film thickness of 1.0 um. This measurement system can detect VOCs
concentrations down to low pptv range.
These instruments to measure $O_3$, $NO_x$ and VOCs concentrations are calibrated carefully.
Detail information for the instruments and the procedures to perform data quality control are
described by Geng et al. (2007), Ran et al. (2009), Tie et al. (2013) and Gao et al. (2017). These
data have been widely used to investigate the diurnal, seasonal and inter-annual variations of $O_3$
in Shanghai (Geng et al., 2007; 2015; Tang et al., 2008; Ran et al., 2009; Gao et al., 2017) and its
chemical mechanism (Geng et al., 2008a; 2008b; Tie et al., 2009a; 2013).
**2.3 WRF-Chem model**
The regional chemical/transport model (Weather Research and Forecasting Chemical model-



WRF-Chem) (Grell et al., 2005) is used to investigate the $O_3$ variations response to emission
changes in Shanghai. The version of the model is improved mainly by Tie et al. (2007) and Li et al.
(2010; 2011). The chemical mechanism chosen in WRF-Chem is the RADM2 (Regional Acid
Deposition Model, version 2) gas-phase chemical mechanism (Stockwell et al., 1990), which
includes 158 reactions among 36 species. The fast radiation transfer module (FTUV) is developed
and used to calculate photolysis rates (Tie et al., 2003), considering the impacts of aerosols and
clouds on the photochemistry (Li et al., 2011). The aerosol modules are developed by EPA CMAQ
(version 4.6) (Binkowski and Roselle, 2003). The wet deposition of chemical species is calculated
using the method in the CMAQ module and the dry deposition parameterization follows Wesely
et al. (1989). The ISORROPIA version 1.7 is used to calculate the inorganic aerosols (Nenes et al.,
1998). The secondary organic aerosol (SOA) is predicted using anon-traditional SOA module,
including the volatility basisset (VBS) modeling approach and SOA contributions from glyoxal and
methylglyoxal. The major physical processes employed in WRF are summarized as the Lin
microphysics scheme (Lin et al., 1983), the Yonsei University (YSU) PBL scheme (Hong et al., 2006),
the Noah Land surface model (Chen and Dudhia, 2001), and the long wave radiation
parameterization (Dudhia, 1989).

The domain is set up to covered a region (centered at 32.5°N, 118°E) of 356×345 grids with
a horizontal resolution of 6 km (Zhou et al., 2017). The initial and lateral boundary conditions of
the meteorology are extracted from the NCEP FNL reanalysis data. The lateral meteorological
boundary is updated every 6 h. The chemical lateral boundary conditions are constrained by the
global chemical transport model (MOART–Model for Ozone and Related chemical Tracers) with
aerosol formation modules (Tie et al., 2001; Emmonset al., 2010). Both the chemical and
dynamical integration step is set as 60 s. The Multi-resolution Emission Inventory for China (MEIC)
developed by Zhang et al. (2009) is used in WRF-Chem for the domain except Shanghai. The
anthropogenic emissions (including CO, $NO_x$, $SO_2$ and VOCs) for Shanghai are developed by Tie et
al. (2013) based on the MIRAGE-shanghai field campaign. $NO_x$ and $SO_2$ emissions in YRD region
are adjusted by Zhou et al. (2017) according to the performance evaluation of WRF-Chem
prediction for about 195 cities during 2014-2015. The biogenic emissions are calculated online
using the MEGAN (Model of Emissions of Gases and Aerosol from Nature) model developed by
Guenther et al. (2006).
**2.4 OZIPR model**
The ozone isopleths diagram for Shanghai is plot by OZIPR (Ozone Isopleths Plotting Package
Research) model (Gery and Crouse, 2002). The OZIPR model employs a trajectory-based air
quality simulation model in conjunction with the Empirical Kinetics Modeling Approach (EKMA)
to relate $O_3$ concentrations levels of organic and nitrogen oxide emissions. OZIPR simulates
complex chemical and physical processes of the lower atmosphere through a trajectory model.
The physical representation is a well-mixed column of air extending from the ground to the top of
the mixed layer. Emissions from the surface are included as the air column passes over different
emission sources, and air from above the column is mixed in as the inversion rises during the day.
$O_3$ precursor concentrations and ambient information such as temperature, relative humidity and
boundary layer height from measurements in Shanghai were specified for each single run.
Therefore a series of simulations were performed to calculate peak $O_3$ concentration as a
function of initial precursor concentrations (Tang et al., 2008; Geng et al., 2008b).






**3 Variability of O₃ and its precursors measured in Shanghai**

**3.1 Variation of O₃ concentration**

Fig. 2a and b show the annual variation of $O_3$ concentration at downtown site XJH and sub-urban site PD respectively from 2006 to 2015. The $O_3$ concentrations increase notably during the past ten years with the increasing rate of 1.057 ppbv yr$^{-1}$ at XJH and 1.346 ppbv yr$^{-1}$ at PD respectively. It is consistent with the reported $O_3$ increasing trend ranging from 1-2 ppbv yr$^{-1}$ at background and urban sites in eastern China during 2001 to 2015 (Tang et al., 2009; Ma et al., 2016; Sun et al., 2016). In 2006, the mean $O_3$ concentrations at XJH and PD are 20 ppbv and 28 ppbv respectively. While in 2017, the mean $O_3$ concentrations at the two sites increase to 35 ppbv and 42 ppbv respectively, with 26% and 30% enhancement compared with that in 2006. The mean $O_3$ concentration at downtown site XJH during 2006 to 2015 is 32 ppbv, which is significantly lower than that at sub-urban site PD of 36 ppbv, suggesting the $O_3$ is depressed in downtown area. Geng et al. (2007) suggested that the $O_3$ production in the city of Shanghai was under VOC-limited regime, thus higher $NO_x$ in downtown resulted in lower $O_3$ concentration. Considering the inhomogeneous spatial distribution of the precursors of $O_3$ in Shanghai (Geng et al. 2008a), we extend the analysis on $O_3$ variations to a broader scope by using the $O_3$ measurements from 31 sites provided by Shanghai Environmental Monitoring Center, covering the entire Shanghai area. It is shown in Fig. 2c that the median of the $O_3$-8h concentration also increases significantly from 2006 to 2015, with the increasing rate of 1.571 ppbv yr$^{-1}$, indicating that the significant increasing trend of $O_3$ concentration not only occurs in the city of Shanghai, but also expanded to a larger area nearby Shanghai. Li et al. (2019) also reported a regional $O_3$ increasing phenomena in summer during 2013 to 2017 from Shanghai to Beijing in eastern China.

In order to analyze the individual contribution to the long-term $O_3$ trend, the variations of $O_3$ precursors, and meteorological parameters are measured and showed in the following sections.

**Figure 2.** The mean annual $O_3$ concentration (ppbv) from 2006 to 2015 at (a) downtown site XJH and (b) sub-urban site PD, both presenting the significant increasing trends with 1.057 ppbv yr$^{-1}$ at XJH and 1.346 ppbv yr$^{-1}$ at PD. The variation of the median 8-h $O_3$ concentration (ppbv) from 2006 to 2015 averaged for 31 sites over Shanghai (c), also shows the increasing variability of 1.571 ppbv yr$^{-1}$.

**3.2 Variations of the precursors (NO$_x$ and VOCs)**

It is well known that the tropospheric $O_3$ formation is throughout a complicated photochemical process, and is strongly related to the precursors of $O_3$ (VOCs and $NO_x$). According to the previous studies (Geng et al., 2007; Ran et al., 2009), the chemical formation of $O_3$ in Shanghai is revealed to be under VOC-limited. Thus either enhancement of VOCs or reduction in $NO_x$ would both result in the growth of $O_3$ concentration. In order to better understanding the factors possibly driving the $O_3$ increasing trend depicted in Fig. 2, the variations of $NO_x$ and VOCs concentrations at XJH and PD in the same period are presented in Fig. 3. The $NO_x$ concentrations present significant decreasing trend from 2006 to 2015 at both XJH and PD sites, which is opposite to the increasing trend of $O_3$ variations in Fig. 2. At XJH, the decreasing rate of $NO_x$ is 2.15 ppbv yr$^{-1}$,



which is more remarkable than that at PD site of 1.86 ppbv yr$^{-1}$. According to the studies by Lin et
al (2017), the reduction of NO$_x$ concentration in Shanghai was likely attributed to the
implementation of stringent emission control strategy for transportation, including improvement
of gas quality, popular usage of electricity cars, and limitation of heavy cars into the urban zones.
These regulations significantly decrease the emissions of NO$_x$ into the atmosphere, resulting in
lower NO$_x$ concentrations. Zheng et al. (2018) also reported the 30% reduction of NO$_x$ emission in
the past 5 years in YRD region. In comparison, the VOCs concentrations at XJH and PD decrease
very slightly during 2006 to 2015. At XJH, the mean VOCs concentration during 2013 to 2015 is
about 20 ppbv, which is some lower than that during 2009 to 2012 of 23 ppbv. At PD, the VOCs
concentration shows strong inter-annual variations, ranging from 16 to 22 ppbv. Generally the
VOCs concentration at the downtown site XJH is higher than that at the sub-urban site PD by 14%.
It is consistent with the studies of Cai et al. (2010), suggesting that about 25% of VOCs is
attributed to the vehicles in shanghai urban zones.

**Figure 3.** The mean annual concentrations (ppbv) of NO$_x$ (dots) and VOCs (bars) from 2006 to
2015 at (a) downtown site XJH and (b) sub-urban site PD respectively. The NO$_x$ concentrations at
XJH and PD both present obvious decreasing trends with -2.1 ppbv yr$^{-1}$ and -1.87 ppbv yr$^{-1}$. While
the VOCs concentrations at both sites present no clear inter-annual trends.

**3.3 Meteorological impacts on O$_3$ photolysis, dispersion and transport**
In addition to the precursors, meteorology such as solar radiation and wind speed and directions
also plays the important roles in O$_3$ concentration through the photochemical and physical
processes. Fig. 4 shows the annual variation of wind speed and total solar radiation from 2006 to
2015. The solar radiation presents weak annual variations ranging from 140 to 150 Wm$^{-2}$,
exhibiting a large variability but without a significant trend. As a result, the variation of solar
radiation cannot explain the significant change of O$_3$ concentration on the view of photolysis. The
wind speed is usually regarded as the indicator for the dispersion capacity for air pollutants.
Several studies reported that the wind speed in winter in eastern China presented decreasing
variability during the past 40 years due to the decadal variation of winter monsoon affecting the
haze occurrence (Wang et al., 2016; Zhao et al., 2016; Xu et al., 2017). While high O$_3$ events
usually occur in summer season for middle-latitude cities such as Shanghai (Wang et al., 2017).
The mean summer wind speeds in Fig. 4a show slight decreasing from 2006 to 2015, while
without significant trends. The wind speed fluctuates between 3.3 ms$^{-1}$ to 3.9 ms$^{-1}$ except the
minimum value in 2014 (2.9 ms$^{-1}$) due to fewer typhoon in the period. Without 2014, the
variability of summer wind speed is insignificant, with a trend of -0.02 m s$^{-1}$ yr$^{-1}$, which could not
be regarded as the dominant factor to interpret the increasing O$_3$ trend. Local O$_3$ concentration
would be affected by transport of upstream plumes usually determined by wind direction. Geng
et al. (2011) suggested that O$_3$ concentration was higher in west wind compared with other wind
sectors in Shanghai indicating the possible O$_3$ transport from western area out of Shanghai. Fig. 5
presents the annual wind rose at Baoshan site from 2006 to 2015, presenting the very similar
pattern of wind direction in each year. The mean wind direction concentrates in the sector
between 60-80 degree, suggesting the dominant wind in Shanghai is easterly accounting for 50%.
The east wind in Shanghai usually carries with the clean air mass from the sea to improve the



local air quality (Xu et al., 2015). The frequency of west wind changes little during 2006 and 2015
ranging from 10-15%, suggesting that the regional transport is not a major factor driving the O₃
increasing. Based on the above analysis, it is speculated that the rapid O₃ increasing during
2006–2015 in shanghai is likely attributed to the reduction of NOₓ concentration as a result of the
VOC-limited condition for O₃ production.
**Figure 4.** The annual variation of (a) summer wind speed (m s$^{-1}$) and (b) total solar radiation (W
m$^{-2}$) from 2006 to 2015 in Shanghai. Both wind speed and the solar radiation present weak
inter-annual variations but without significant trends.
**Figure 5.** The wind rose in each year from 2006 to 2015 in Shanghai. The red line means the
resultant vector suggesting the dominant wind direction.
**3.4 Different O₃ variability in nighttime and daytime**
To further qualify the changes of O₃ precursors, especially NOₓ on the measured O₃ variability,
intensive model studies are applied. At first, a brief O₃ daytime and nighttime chemistry is
described. As we know, O₃ in the Earth's atmosphere is ultimately formed from the combination
reaction of atomic oxygen (O$^3$P) and molecular oxygen (O₂) (R1). In the troposphere with little UV
radiation, photolysis of NO₂ at wavelengths ⩽424 nm (R2) is the primary source of O$^3$P atoms
and prompts O₃ production. Once formed, O₃ readily lost with reaction with NO to converts back
to NO₂ (R3). The (R1-R3) reactions result in a null cycle when no other chemical species are
involved. However, in reality, the troposphere contains alternative oxidants (i.e., HO₂ and RO₂)
that efficiently convert NO to NO₂ (R4 and R5), resulting in the accumulation of O₃.
$$O(^3P) + O_2 + M \rightarrow O_3 + M \qquad (R1)$$
$$NO_2 + h\nu \rightarrow NO + O(^3P) \qquad (R2)$$
$$O_3 + NO \rightarrow NO_2 + O_2 \qquad (R3)$$
$$HO_2 + NO \rightarrow OH + NO_2 \qquad (R4)$$
$$RO_2 + NO \rightarrow RO + NO_2 \qquad (R5)$$
The O₃ chemical mechanism in daytime includes both production and loss processes. In
contrast, in nighttime, the photochemical production ceases, and there mainly exists loss process
for O₃. Fig. 6 shows the O₃ variations in daytime and nighttime respectively from 2006 to 2015 at
XJH and PD sites. Both daytime and nighttime O₃ concentrations present significant increasing
trend at two sites, which is consistent with the results in Fig. 2. It is worthy to note that O₃
concentration in nighttime increases more rapidly than that in daytime. For example, at XJH the
nighttime O₃ concentration increases at the rate of 1.47 ppbv yr$^{-1}$ from 2006 to 2015, higher than
that in daytime of 0.85 ppbv yr$^{-1}$. At PD, the increasing rate of O₃ concentration is 1.22 ppbv yr$^{-1}$
in nighttime, also higher than that in daytime of 0.91 ppbv yr$^{-1}$. These results suggest that the
reduction of NOₓ concentration from 2006 to 2015 has different effects on daytime and nighttime
O₃ variations. The O₃ concentration in nighttime is more sensitive to NOₓ reduction, resulting in
less O₃ lost compared with that in daytime. The results in Fig. 6 also show that the increasing rate
of nighttime O₃ in downtown site XJH is higher than that at sub-urban site PD due to the more
reduction of NOₓ concentration in downtown area.




**Figure 6.** The annual variations of daytime and nighttime $O_3$ concentration (ppbv) from 2006 to
2015 at (a) downtown site XJH and (b) sub-urban site PD.

**4 WRF-Chem study on the $O_3$ variation response to emission change**
**4.1 Design of the model experiments scheme**
To better understand the role of $NO_x$ emission reduction in $O_3$ variation, the WRF-Chem model is
utilized to calculate the changes of $O_3$ concentrations. Lin et al. (2017) suggested that the $NO_x$
emission was reduced in Shanghai in recent years resulted from the implementation of the
Shanghai Clean Air Action Plan. The $NO_x$ emission in 2015 is estimated at $33.4 \times 10^4$ ton in
Shanghai, reduced significantly by 30% compared with that in 2009 of $44.9 \times 10^4$ ton. Thus it
provided the good opportunity to examine the $O_3$ variation response to the reduction of $NO_x$
emission in Shanghai. The $NO_x$ emissions in 2009 and 2015 are put into WRF-Chem model
respectively to calculate the $O_3$ concentration. The other emissions (including gas and particulate
matter) and meteorology used in WRF-Chem are set same. As a result, the difference of $O_3$
concentrations calculated by WRF-Chem is solely attributed to the change of $NO_x$ emission
between 2009 and 2015, which is furthermore compared with the measurements.
The MIRAGE-shanghai field campaign was conducted in September of 2009 to explore the
$O_3$ chemical formation and transformation in Shanghai (Tie et al., 2013). The mean temperature,
mean wind speed and total precipitation in this month is 25 $^oC$, 2.85 m s$^{-1}$ and 89.5 mm
respectively, which is very close to the climatological conditions during the past ten years from
2006 to 2015, with 24.7 $^oC$ for mean temperature, 2.81 m s$^{-1}$ for mean wind speed, and 126 mm
for total precipitation respectively. In addition, Shanghai is located at the typical sub-tropical area.
The meteorology in September is characterized as the low cloud cover and rain occurrence, the
slight wind speed and humidity, as well as the moderate solar radiation intensity. As suggested by
Tie et al. (2013), the chemical age of $O_3$ plume in Shanghai urban area in September of 2009 was
very young, indicating that the $O_3$ production was more dependent on the local emissions under
such kind of meteorology, hence providing more insights into the $O_3$ chemical mechanism
response to the local emission changes. We chose the meteorology in September of 2009 as the
atmospheric background for the following sensitive experiments by WRF-Chem.
Tie et al. (2009a; 2013) highlighted that the WRF-Chem model was capable of studying the
chemical and physical processed of $O_3$ in September of 2009 during the MIRAGE-Shanghai
campaign. The calculated $O_3$, $NO_x$, VOCs and aerosols by WRF-Chem in clean and polluted
episodes are fairly in agreement with the measurements except HONO, suggesting that the
emission inventory in 2009 used in the model is reasonable for the Shanghai region. Moreover
the VOCs emission in Shanghai is greatly improved according to the measurements from the
MIRAGE-shanghai field campaign by Tie et al. (2013). Such emission from Tie et al. (2013)
representing 2009 scenario is used in this study to conduct the control experiment (T1) as the
baseline to simulate the $O_3$ and $NO_x$ concentrations in September of 2009. The T1 experiment is
composed of 30 model runs for each day in September of 2009. Each model run is initiated at the
20:00 (LST) and performed for 52 h integrations. The first 28 h integration is regarded as model
spin-up periods, the results from the later 24 h integration is captured hourly and averaged for
mean daily concentration of $O_3$ and $NO_x$. The aim of the T1 experiment is to further evaluate the



reliability of the emission inventory in 2009 used in WRF-Chem by fully comparing the calculated
O₃ and NOₓ concentrations with in-situ measurements of 6 sites over Shanghai.

**4.2 The NOx emission in 2009 used for base experiment**

Fig. 7 showed the distribution of NOₓ emission of 2009 scenario (Tie et al., 2013) in Shanghai
used in WRF-Chem model. The NOₓ emission is mostly distributed in the urban zones, suggesting
that transportation is the important source. The NOₓ is largely exported in downtown and two
neighboring sub-urban zones in the east and north respectively. The maximum NOₓ emission is
estimated at 16 kg hr⁻¹ km⁻² at downtown, compared with 2-6 kg hr⁻¹ km⁻² in the sub-urban area.
In addition, there is a small town located in the south of Shanghai with the similar intensity of
NOₓ emission as the sub-urban zones. The total NOₓ emission of 2009 scenario in Shanghai (Fig. 7)
is estimated at $41.4 \times 10^4$ ton in the model, which is close to the $47.8 \times 10^4$ ton suggested by Lin
et al. (2017) according to the Shanghai Environmental Year Book.

**Figure 7.** The distribution of NOₓ emission (kg km⁻² h⁻¹) in 2009 in Shanghai.

**4.3 Performance evaluation on the base experiment**

The mean monthly O₃ concentration in September 2009 is calculated by WRF-Chem and
compared with measurements over 6 sites in Shanghai. It is shown in Fig. 8 that both model
simulations and in-situ measurements highlight the lower O₃ concentration in urban zones than
that in suburb. The simulated O₃ concentration in downtown is 22-24 ppbv, significantly lower
than that at sub-urban (30-35 ppbv) and rural area (40 ppbv), which is consistent with the
measurements. The measured O₃ concentration at downtown site XJH is 22 ppbv, lower than that
at sub-urban site PD and remote site DT by 12 ppbv and 26 ppbv respectively. Geng et al. (2007)
suggested that under VOC-limited regime, the lower O₃ concentration in downtown was resulted
from the higher NOₓ emission, which depressed the O₃ production process. Under high NOₓ
conditions, the OH radicals are lost by the reaction of $NO_2 + OH \rightarrow HNO_3$ (Sillman, 1995). As
a result, higher NOₓ concentration in urban area leads to lower OH concentration, which results
in smaller O₃ production. Tang et al. (2008) also suggested that the O₃ concentration in Shanghai
downtown was higher at weekends than that on weekdays due to the reduced NOₓ concentration.
However the discrepancy is also evident between model results and measurements. For example,
the modeled O₃ concentrations at XJH and PD are about 2-3 ppbv higher than the measurements,
perhaps due to the uncertainty of NOₓ and VOCs emission in urban area suggested by Tie et al.
(2009a). In addition, the calculated O₃ concentrations in the remote site DT and chemical site JS
are lower than measurements by 8 and 5 ppbv respectively. The former is resulted from the
overestimation of the wind speed by WRF-Chem model leading to excessive O₃ transport for
underestimation (Zhou et al., 2017). While the latter is mainly due to the prominent
underestimation of the VOCs emission in the chemical zones suggested by Tie et al. (2009a).

**Figure 8.** The calculated distribution of O₃ concentration by WRF-Chem (shade) in September of
2009 compared with measurements (circles) of 6 sites over Shanghai.

Fig. 9a and b show the daily variations of O₃ and NOₓ concentrations compared between
WRF-Chem simulations and the in-situ measurements over 5 sites. The statistical analysis of
model performance for $O_3$ and $NO_x$ is listed in Table 1 and Table 2 respectively. The calculated
magnitude and daily variation of $O_3$ concentrations agree well with the measurements,
suggesting that both meteorology and photochemistry are well reproduced by WRF-Chem model.
For example, the Root Mean Square Error (RMSE) calculated between modeled and measured $O_3$
concentration are 7.4, 10.5, 12, 8.6, 9.2 ppbv for XJH, JS, DT, PD and BS respectively, and the
difference between the simulation results and in-situ measurement is below 10%, which are very
satisfactory compare with the similar works by Geng et al (2007) and Tie et al. (2013). The
correlated coefficients (R) for the mean daily $O_3$ concentration range from 0.6 to 0.8 above 99%
confidence over 5 sites, indicating good consistency of day by day variations between the model
results and measurements. Comparably the $O_3$ concentration is best simulated by WRF-Chem at
the downtown site XJH and sub-urban site PD with the lower RMSE and better R. However the
discrepancy of daily $O_3$ concentration between the model and measurements is also evident. For
example, a rapid change of $O_3$ concentration from 16 to 19 in September was observed over all
sites, indicating it's a regional event instead of a local phenomenon. The $O_3$ concentration firstly
increases significantly during 16-19 (episode-1), then sharply decreased during the later 4 days
(episode-2). The similar rapid $O_3$ change in Shanghai was also reported by Tie et al. (2009a), and
their explanation is that this episode was mainly related to the intensity of the sub-tropical
high-pressure system on Pacific Ocean in summer. The model captures the $O_3$ variations and
magnitudes during the both risen and fallen episodes very well at downtown site XJH, but
substantially underestimates the increasing variability of $O_3$ concentration during episode-1 at
sub-urban and rural sites by 10-15 ppbv. Geng et al. (2008a) suggested the "chemical transport of
$O_3$" from Shanghai downtown area to the distance of 18-36 km far away, which increased the $O_3$
concentration at sub-urban or rural sites. This "chemical transport of $O_3$" is difficult to be
reflected by WRF-Chem model due to the current inventory is too coarse to accurately reflect the
detailed distribution and variation of $NO_x$ emission, e.g. the $NO_x$ emission from mobile source in
the city. In addition, the underestimation of the $O_3$ concentration at suburb of Shanghai in
summer is possibly attributed to the model bias of sea breeze simulations. Under the condition of
weak sub-tropical pressure, the sea breeze develops at noontime to yield a cycling wind pattern
in Shanghai, leading to the rapid accumulation of high $O_3$ concentration. The WRF-Chem usually
underestimates the sea surface temperature, which tends to accelerate the sea breeze
development and weak the $O_3$ trapping in the city (Tie et al., 2009a). The calculated daily $NO_x$
concentration by WRF-Chem compared with measurements are shown in Fig. 9b. Both the
modeled and measured $NO_x$ concentrations at the remote site DT are very low, with the average
of 1.4 and 2.9 ppbv respectively due to seldom anthropogenic emissions there. The calculated
$NO_x$ concentration at XJH and PD are generally well consistent with the measurements with the
excellent R of 0.8 and 0.82 and small RMSE of 6.9 and 7.5 ppbv respectively. However the $NO_x$
concentration is underestimated by WRF-Chem at sub-urban site BS in the steel zone. The
calculated $NO_x$ concentration at BS is 16.1 ppbv, which is lower than the measurements by 5 ppbv.
The difference of $NO_x$ concentrations between the model and observations is generally above
10%, suggesting the performance of $NO_x$ simulation is somewhat lower than that of $O_3$. It was
also reported by Tie et al. (2007; 2009b; 2013), during the evaluation of the $NO_x$ calculations by
WRF-Chem in MIRAGE-Shanghai and MIRAGE-mex campaign studies. The lifetime of $NO_x$ at the
surface is about 1-2 days, shorter than $O_3$. Thus the $NO_x$ concentration is determined by the



detailed emissions and dynamical factors, which need to develop the advanced inventory with
higher resolution to reproduce both the spatial distributions and temporal variations of $NO_x$
emission.
**Figure 9.** The calculated mean daily concentrations (ppbv) of (a) $O_3$ and (b) $NO_x$ at 5 sites in
September of 2009 by WRF-Chem (red circles) and compared with measurements (blue circles).

**4.4 Sensitive study on the $O_3$ variability response to the emission change**
The T1 experiment shows the excellent performance for $O_3$ and $NO_x$ simulations, including the
spatial distribution pattern, and the day by day variation and magnitude. It is indicated that the
emission in 2009 scenario used in WRF-Chem is reasonable, and the model is efficient for
conducting the sensitive studies on $O_3$ variation response to the emission change. In order to
better understand the measured long-term trend of $O_3$ concentration during the past ten years in
Shanghai and its relationship to the emission reduction, several sensitive studies are conducted in
this study (Table 3). The control study of T1 is conducted based on the $NO_x$ emission in 2009
scenario in Shanghai. According to the study of Lin et al. (2017), the $NO_x$ emission in 2015 in
Shanghai is reduced by 30% compared with that in 2009. Thus we conduct the sensitive
experiment T2 by WRF-Chem, cutting the $NO_x$ emission by 30% compared with T1, whereas
keeping the other emissions and meteorology same as T1. As a result, the calculated $O_3$
difference between T1 and T2 is likely attributed to the $NO_x$ emission reduction between 2015
and 2009.
Fig. 10a shows the distribution of the difference of $O_3$ concentration simulated by T1 and T2
(T2-T1). The reduction of $NO_x$ emission has the obvious effect on the magnitude and distribution
of $O_3$ concentration. The $O_3$ concentration increases notably in urban area corresponding to the
higher $NO_x$ emissions in Fig. 7, ranging from 2-7 ppbv. The enhancement of $O_3$ concentration is
most significant in downtown and neighboring sub-urban zones, as well as the southern town,
generally more than 4 ppbv. For example, the maximum increase in $O_3$ concentration is 6.4 ppbv
occurred at downtown site XJH, followed by 4-5 ppbv at sub-urban site PD. The increasing rates
of $O_3$ trend at XJH and PD are estimated at 1.06 ppbv $yr^{-1}$ and 0.96 ppbv $yr^{-1}$ from 2009 to 2015
by WRF-Chem, which is consistent to the observed $O_3$ growth variability (Fig. 2) of 1-1.3 ppbv $yr^{-1}$.
The response of $O_3$ concentration to the $NO_x$ reduction is not evident in the rural area including
the eastern part of Shanghai and the island with low $NO_x$ emissions. The comparison of T1 and T2
further illustrates the speculation that the significant increasing trend of $O_3$ concentration during
the past ten years in Shanghai is mostly attributed to the reduction of $NO_x$ emission as a result of
the implementation of Shanghai Clean Air Action Plan.
The $O_3$ chemical formation is strongly related to $NO_x$ and VOCs concentrations. As discussed
by Geng et al. (2008a) the $O_3$ chemical formation is clearly under VOC-limited regime in Shanghai
and its neighboring area. Under the high $NO_x$ condition, NO tends to react with $O_3$ instead of $NO_2$,
flowing by $NO_2 + OH \rightarrow HNO_3$, causing the decrease of the reactivity and ensuing $O_3$
concentrations. Thus reduced $NO_x$ emission would result in increase in $O_3$ concentration, which
has been discussed in Fig. 10a.
Despite of minor change of VOCs in the last ten years, it is worth to investigate the effect of
the VOCs changes on $O_3$ concentrations in Shanghai. For this purpose, we conduct a sensitive



study (T3), with 50% increase of VOCs emission compared with T1, but keeping $NO_x$ and other
emissions as well as the meteorology same as T1. For RADM2 gas mechanism used in WRF-Chem,
the VOCs are surrogated into 14 species, such as alkane, alkene, aromatic, formaldehyde, etc. All
the species of VOCs are increased by 50% at every model grid over Shanghai and at every hour.
The difference of $O_3$ concentration between T3 and T1 (T3-T1) is shown in Fig. 10b. As we
expected, the $O_3$ concentration in Shanghai is sensitive to the enhancement of VOCs emission,
increased by 3-4 ppbv in urban area due to more NO is converted to $NO_2$ by reaction with $RO_2$
and $HO_2$. Furthermore, the abundant $O_3$ plumes produced in the urban zones significantly
transport to the downwind areas about 100-200 km away, resulting in elevated $O_3$ concentration
in the western Shanghai by about 2 ppbv. According to Tie et al. (2013), the $O_3$ plume released in
Shanghai urban area can be transported to downwind of the city by about 100-150 km away in
the MIRAGE-shanghai field campaign. The model studies of T1, T2 and T3 highlight that under the
emission of 2009 scenario, the $O_3$ chemical production is clearly under VOC-limit regime, either
decreasing $NO_x$ concentration or increasing VOCs concentration would result in the $O_3$
enhancement. The analysis on in-situ measurements and model experiments jointly suggests that
the significant $O_3$ increasing trend during the past ten years in Shanghai is mainly attributed to
the large reduction of $NO_x$ emission.

**Figure 10.** The difference of O3 concentration (ppbv) between (a) T2 and T1 (T2-T1), (b) T3 and
T1 (T3-T1) respectively conducted by WRF-Chem model. The difference between T2 and T1 lies in
the $NO_x$ emissions set in T2 (2015 scenario) is 30% lower than that in T1 (2009 scenario), which is
estimated by Lin et al. (2017) according to the Shanghai Environment Yearbook. The difference
between T3 and T1 is dependent on that the VOCs emission in T3 is 50% higher than that in T1.

**4.5 The variation of $O_3$ production regime response to emission change**
The $O_3$ chemical mechanism in Shanghai was explored by several studies based on the in-situ
measurements around 2008 and 2009. Geng et al. (2008a; 2008b), Ran et al. (2009) and Tie et al.
(2009a) all revealed that the $O_3$ production around 2008 and 2009 in Shanghai was clearly under
VOC-limit regime which was further illustrated by the above model studies. As indicated in Fig. 3,
the significant decrease of $NO_x$ concentration is observed from 2009 to 2015 in Shanghai, while
the VOCs concentration changed little during the same period. As we know, the $O_3$ chemical
formation is strongly related to $NO_x$ and VOCs concentrations with nonlinearity. Thus the
different variability of $NO_x$ and VOCs concentration from 2009 to 2015 inevitably has the large
effect on the $O_3$ production regime, which need to be investigated deeply.
The complex relationship among $NO_x$, VOCs and $O_3$ concentrations is usually depicted by $O_3$
isopleths diagram. The $O_3$ isopleths plot (Fig. 11) in Shanghai used in this study is constructed by
the OZIPR model based on the in-situ measurements of $O_3$, $NO_x$, VOCs and meteorology. Under
high VOCs and low $NO_x$ condition (low $NO_x$/VOCs ratio), the $O_3$ production is not sensitive to
VOCs, while positively correlated to $NO_x$ concentration, which is viewed as $NO_x$-limited regime. By
contrast, under low VOCs and high $NO_x$ condition (high $NO_x$/VOCs ratio), the $O_3$ production tends
to increase with the VOCs growth or $NO_x$ reduction, which is regarded as VOC-limited regime. The
$NO_x$-limited and VOC-limited regime is divided by a ridge line (the dot-dash line in Fig. 11) in the
$O_3$ isopleths plot. The $O_3$ production is not sensitive to neither $NO_x$ concentration nor VOCs



concentration when near the ridge line, which is regarded as the transition regime.

The $O_3$ chemical production regime at XJH and PD in 2009 and 2015 is positioned

respectively in Fig. 11. In 2009 the $O_3$ production at both XJH and PD sites (marked as red and
blue hollow circle respectively) are clearly under VOC-limited regime. Thus decrease in $NO_x$
concentration leads to the $O_3$ enhancement, which is highlighted by the previous in-situ
measurements and model experiments. Since then the $O_3$ production regime tends to move
toward the dot-dash line due to the significant reduction of $NO_x$ concentration accompanied with
the relative less change of VOCs at the two sites. In 2015 the $O_3$ production at XJH (marked as red
solid circle) is still under VOC-limited regime, but for PD (marked as blue solid circle), it is close to
the dot-dash line, approaching the transition regime between VOC-limited to $NO_x$-limited. This
result suggests that if the $NO_x$ emission keeps reduction after 2015 assuming the VOCs
concentration keeps constant, the $O_3$ concentration will continue to increase at XJH, while at PD
the $O_3$ concentration is supposed to be insensitive to the $NO_x$ change. According to the $O_3$
chemical regime depicted in Fig. 11, if the $NO_x$ concentration decreases by 5 ppbv after 2015, the
peak $O_3$ concentration at XJH will further increase by 3 ppbv, whereas at PD it seems to change
very slightly. To better understand this further change, more sensitive studies of WRF-Chem are
conducted in the following sections.

**Figure 11.** The $O_3$ chemical production at downtown site XJH and sub-urban site PD in 2009 and
2015 depicted by $O_3$ isopleths diagram. The hollow and solid red circles denote $O_3$ production
regime at XJH in 2005 and 2019 respectively. The hollow and solid blue circles denote $O_3$
production regime at PD in 2005 and 2019 respectively

**5 The future $O_3$ evaluation**
**5.1 The $O_3$ level in 2020**
According to the Shanghai Clean Air Action Plan, the $NO_x$ emission in Shanghai will be further
reduced by 20% in 2020 compared with that in 2015. According to the above analysis based on
the $O_3$ isopleths plot (Fig. 11), the $O_3$ concentrations in downtown and sub-urban seem to have
distinct different responses to further $NO_x$ reduction after 2015. In order to better understand
the future $O_3$ variation, the sensitive experiment T4 is conducted by WRF-Chem with 20%
reduction of $NO_x$ emission compared with T2. T2 and T4 represent the $NO_x$ emission in 2015 and
2020 respectively. The other emissions and meteorology are set to be same as T1. The difference
of $O_3$ concentration between T2 and T4 (T4-T2) is presented in Fig. 12a. The $O_3$ concentration
keeps increasing in downtown area such as XJH site, ranging from 2-4 ppbv. However, for the
sub-urban zones such as the PD site, the $O_3$ concentration changes very little response to the
further $NO_x$ reduction, ranging from 0-1 ppbv. As discussed in Fig. 11, in 2015 the $O_3$ production
at PD is possibly under the transition regime from VOC-limited to $NO_x$-limited near the ridge line.
As a result, the $O_3$ concentration is not sensitive to the variation of $NO_x$ concentration. However
the $O_3$ concentration in the suburb zones generally decreases by 1ppbv, indicating that with the
further $NO_x$ reduction after 2015 the $O_3$ chemical production transfers from VOCs-limited to
$NO_x$-limited regime in the rural of Shanghai.

It is suggested in Fig.11 that the $O_3$ production at downtown site XJH in 2015 is still under

VOC-limited regime despite of the significant $NO_x$ reduction. The $O_3$ concentration would be also



sensitive to the variation of VOCs concentration. Thus the sensitive experiment T5 is conducted
by WRF-Chem model with 50% enhancement of VOCs emission compared with T2 (representing
the emission in 2015 scenario). It is presented in Fig. 12b that the $O_3$ concentration increases by
2-3 ppbv in downtown area due to the enhancement of VOCs, suggesting that the $O_3$ production
at downtown in 2015 is still under VOC-limited regime, which is consistent with the results in Fig.
11. Moreover the $O_3$ plumes produced in urban area transport to the downwind area to
accumulate the high $O_3$ concentration in the western area to Shanghai by 2 ppbv. While at
sub-urban site PD, the $O_3$ concentration changes less than1 ppbv response to the increase in
VOCs emission, which is similar as the very weak $O_3$ variations relative to the $NO_x$ reduction in Fig.
12a. Overall, the models studies of T4 and T5 jointly suggest that the $O_3$ concentration at
sub-urban site PD in 2015 is not sensitive to either $NO_x$ or VOCs variations due to the $O_3$
production is under the transition regime depicted in the $O_3$ isopleths plot.
**Figure 12.** The difference of O3 concentration (ppbv) between (a) T4 and T2 (T4-T2), (b) T5 and
T2 (T5-T2) respectively conducted by WRF-Chem model. The difference between T4 and T2 is
that the $NO_x$ emissions set in T4 (2020 scenario) is 20% lower than that in T2 (2015 scenario),
which is estimated according to the Shanghai Clean Air Action Plan. The difference between T5
and T2 lies in that the VOCs emission in T5 is 50% higher than that in T2.
**5.2 The $O_3$ chemical production after 2020**
The above study shows that the $O_3$ production at sub-urban site PD in 2020 will likely transfer
from VOCs-limited regime to $NO_x$-limited regime without consideration of possible VOCs changes.
For the purpose of the $O_3$ pollution control strategy, it is worth to estimate the $O_3$ level response
to emission change after 2020 in Shanghai. It is also essential to access how many $NO_x$ emission
need to be cut after 2020 will cease the $O_3$ enhancement in downtown area. Thus the sensitive
experiment T6 is conducted by further 20% reduction of $NO_x$ emission from 2020 scenario (T4).
The difference of $O_3$ concentration between T6 and T4 (T6-T4) is shown in Fig. 13a. It is clear that
the $O_3$ concentration at downtown keeps nearly constant regardless of the further reduction of
$NO_x$ emission after 2020. That is to say the increasing trend of $O_3$ in downtown with the $NO_x$
reduction ceases after 2020, indicating that the $O_3$ production likely approachs the transition
regime. In addition, the $O_3$ concentration decreases significantly out of the downtown area,
ranging from 2-3 ppbv in sub-urban zones, and more than 4 ppbv in suburb, indicating that the
$O_3$ production in Shanghai transfers to $NO_x$-limited regime after 2020 except the downtown area
where the $O_3$ production is likely near the transition zone. On the other hand, if the $NO_x$ emission
is kept constant after 2020 as T4, while the VOCs emission is increased by 50% conducted in T7
experiment, the $O_3$ concentration (Fig. 13b) changes little in both urban and suburb area in
Shanghai which is different from the previous model study of T5 the T3 when $O_3$ production is
under VOC-limited condition. It is suggested that the $O_3$ concentration after 2020 is not sensitive
to the variation of VOCs concentration because the continuous reduction of $NO_x$ emission keeps
in promoting the $O_3$ production to transfer into $NO_x$-limited regime. Thus further reduction of
$NO_x$ tends to decrease the $O_3$ concentration in Shanghai.
**Figure 13.** The difference of O3 concentration (ppbv) between (a) T6 and T4 (T6-T4), (b) T7 and



659 T4 (T7-T4) respectively conducted by WRF-Chem model. The $NO_x$ emissions set in T6 is 20% lower
660 than that in T4 (2020 scenario). The VOCs emission in T7 is 50% higher than that in T4.


662 **Conclusions**

663 $O_3$ pollution is a serious issue in China. Better understanding the elevated $O_3$ and its response to
664 emission change is important for Chinese megacities. In this study, we analyze the increasing
665 trend of $O_3$ concentration by long-term measurements of $O_3$ and its precursors as well as
666 meteorology in Shanghai combined with the WRF-Chem model. The $O_3$ production regime
667 response to the emission change in Shanghai during the past ten years is also explored by $O_3$
668 isopleths plot. In addition, the future $O_3$ variation and its chemical production in Shanghai are
669 evaluated by WRF-Chem model. The main conclusions are summarized as follows:

670  (1) The $O_3$ concentration measured in Shanghai increased significantly from 2006 to 2015
671 with the rate of 1.057 ppbv $yr^{-1}$ at downtown site XJH and 1.346 ppbv $yr^{-1}$ at sub-urban site PD
672 respectively. The observed increasing trend of $O_3$ is not limited in the urban zones but expanded
673 to the larger scale covering the total Shanghai city. The $NO_x$ and VOCs concentrations presented
674 different variability from $O_3$ during the same period, in which $NO_x$ concentration decreases
675 significantly at both XJH and PD sites, whereas the VOCs changes very little without evident
676 trend.

677  (2) Because there are minor trends of measured $O_3$ photolysis, local dispersion and regional
678 transport resulted from meteorology, it is speculated that the significant $O_3$ increasing trend
679 during 2006 to 2015 in Shanghai is likely attributed to the reduction of $NO_x$ concentration as a
680 result of the strong VOCs-limited regime for $O_3$ production. The nighttime $O_3$ is more sensitive to
681 $NO_x$ reduction than that in daytime, because of more $O_3$ is depressed by $NO_x$ in nighttime. As a
682 result, the observed nighttime $O_3$ concentration at XJH and PD increases more rapidly than that
683 in daytime response the $NO_x$ reduction.

684  (3) The WRF-Chem model is utilized to calculate the long term $O_3$ variations response to
685 emission change. The sensitive experiments illustrate that either reduction of $NO_x$ emission or
686 growth of VOCs emission conducted by WRF-Chem lead to the significant enhancement in $O_3$
687 concentration in urban zones in 2009 as the baseline, indicating the $O_3$ production is clearly
688 under VOC-limited regime. The calculated $O_3$ concentration increases by 1-7 ppbv in urban zones
689 from 2009 to 2015 resulted from 30% reduction of $NO_x$ emission estimated by Shanghai
690 Environmental Monitoring Center. The enhancement of $O_3$ concentration is significant in urban
691 zones generally more than 4 ppbv, with the maximum elevation of 6-7 ppbv occurred at
692 downtown area, which is consistent with the measurements. The increasing rates of $O_3$ trend at
693 downtown site XJH and sub-urban site PD are estimated at 1.06 ppbv $yr^{-1}$ and 0.96 ppbv $y^{-1}$ from
694 2009 to 2015 by WRF-Chem, which is close to the observed $O_3$ growth variability of 1-1.3 ppbv
695 $yr^{-1}$. This result suggests that the observed increasing trend of $O_3$ concentration during the past
696 ten years in Shanghai is likely attributed to the reduction of $NO_x$ emission under the VOC-limited
697 condition for $O_3$ production.

698  (4) The model sensitive study suggests that significant decrease in $NO_x$ concentration
699 combined with the obscure VOCs variation from 2006 to 2015 gradually promotes the $O_3$
700 chemical production in Shanghai from VOC-limited to $NO_x$-limited, which is consistent with the $O_3$
701 isopleths diagram. The $O_3$ isopleths plot shows that $O_3$ production is in VOC-limited regime in
702 both downtown site XJH and sub-urban site PD in 2009. With the 30% reduction of $NO_x$ emission



from 2009 to 2015 estimated by Shanghai Environmental Monitoring Center, the $O_3$ production in
XJH is still under VOC-limited regime, while the $O_3$ production moves to the transition regime in
PD, suggesting that the $O_3$ concentration in sub-urban zones is not sensitive to the variation of
either $NO_x$ or VOCs concentration.

(5) In order to better understand the $O_3$ control strategy in Shanghai, the future $O_3$
production is estimated by WRF-Chem. The $O_3$ concentration in Shanghai downtown would keep
increasing till 2020 with the 20% reduction of $NO_x$ emission after 2015 estimated by Shanghai
Clean Air Action Plan. If the $NO_x$ emission is further decreased by 20% after 2020, The $O_3$
concentration will decrease by 2-3 ppbv in sub-urban zones, and more than 4 ppbv in suburb.
While the $O_3$ concentration in downtown is not sensitive to either $NO_x$ reduction or VOCs
enhancement after 2020, indicating the $O_3$ production in shanghai will transfer to $NO_x$-limited
regimes except downtown where the $O_3$ production is likely close to the transition regime.
Further reduction of $NO_x$ emission after 2020 tend to mitigate the $O_3$ pollution in Shanghai.

*Data availability*. The data used in this paper can be provided upon request from Jianming Xu
(metxujm@163.com).

*Author contributions*. XT came up with the original idea of investigating the impact of emission
change on long term O3 variations by. XT and JX designed the analysis method. JX conducted the
analysis. WG, YL and QF provided the observational data and helped in discussion.

*Competing interests*. The authors declare that they have no conflict of interest.

*Acknowledgements.* This study was funded by the National Key R&D Program of China (grant
2018YFC0213800), the National Natural Science Foundation of China (91644223, 41430424 and
41730108).





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





**Table 1.** Statistical analysis on $O_3$ simulation in September of 2009 by WRF-Chem model compared with measurements of 5 sites (XJH, JS, DT, PD, BS) over Shanghai. MO and MM represent the mean value (unit: ppbv) of observed and modeled $O_3$ concentration respectively. RMSE and R are the Root Mean Square Error and correlated coefficient respectively calculated between modeled and measured $O_3$ concentration.

|  | MO | MM | RMSE | R (99% confidence) |
|---|---|---|---|---|
|  | | ppbv | | \ |
| XJH | 21.6 | 23.0 | 7.2 | 0.78 |
| JS | 34.6 | 30.0 | 10.3 | 0.64 |
| DT | 47.3 | 40.3 | 12.0 | 0.61 |
| PD | 33.5 | 34.9 | 8.6 | 0.74 |
| BS | 31.7 | 31.2 | 9.3 | 0.67 |

**Table 2.** Statistical analysis on $NO_x$ simulation in September of 2009 by WRF-Chem model compared with measurements of 5 sites (XJH, JS, DT, PD, BS) over Shanghai. MO and MM represent the mean value (unit: ppbv) of observed and modeled $NO_x$ concentration respectively. RMSE and R are the Root Mean Square Error and correlated coefficient respectively calculated between modeled and measured $NO_x$ concentration.

|  | MO | MM | RMSE | R (99% confidence) |
|---|---|---|---|---|
|  | | ppbv | | \ |
| XJH | 32.1 | 33.7 | 7.0 | 0.74 |
| JS | 14.9 | 14.7 | 7.6 | 0.61 |
| DT | 3.0 | 1.5 | 2.3 | 0.6 |
| PD | 20.3 | 16.8 | 7.5 | 0.82 |
| BS | 21.6 | 16.1 | 9.8 | 0.8 |

**Table 3.** Scheme of WRF-Chem sensitivity simulations.

| Simulation | $NO_x$ EI | VOCs EI | Meteorology |
|---|---|---|---|
| T1 (Control Run) | 2009 | 2009 | September of 2009 |
| T2 | 2015 (30% reduction) | 2009 | September of 2009 |
| T3 | 2009 | 50% increasing | September of 2009 |
| T4 | 2020 (50% reduction) | 2009 | September of 2009 |
| T5 | 2015 | 50% increasing | September of 2009 |
| T6 | 70% reduction | 2009 | September of 2009 |
| T7 | 2020 (50% reduction) | 50% increasing | September of 2009 |






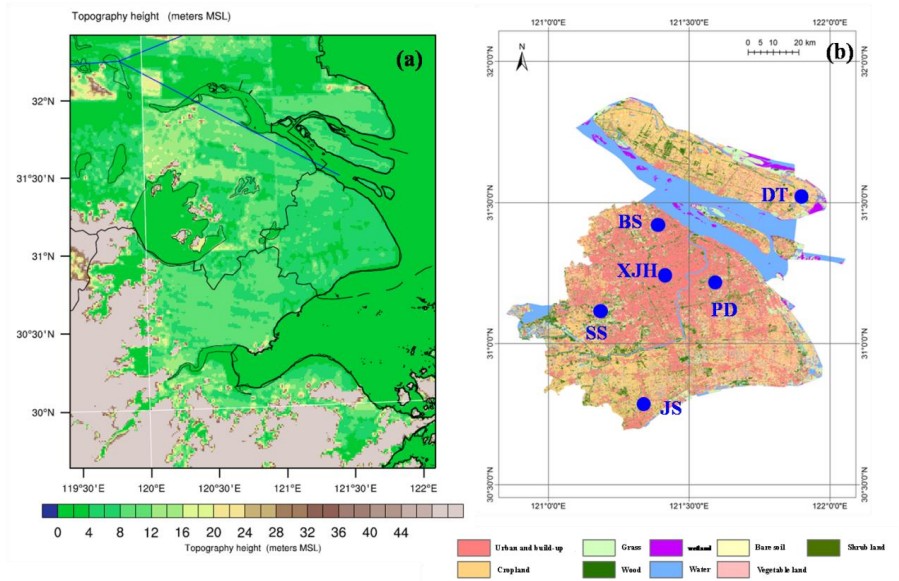


**Figure 1.** (a) The distribution of topography height in Shanghai and its neighboring area. (b) The
distribution of land-use category in Shanghai. The locations of the 6 sites (XJH, BS, PD, SS, JS, DT)
are described by blue dots.




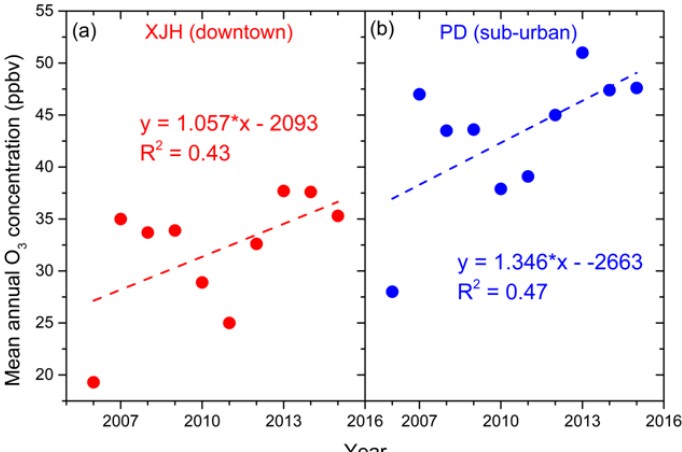


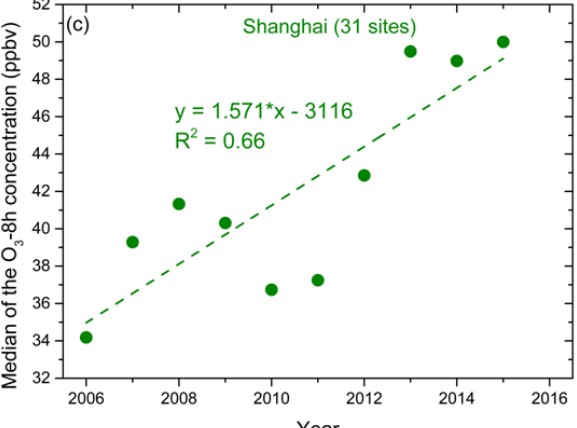


**Figure 2.** The mean annual $O_3$ concentration (ppbv) from 2006 to 2015 at (a) downtown site XJH
and (b) sub-urban site PD, both presenting the significant increasing trends with 1.057 ppbv yr$^{-1}$
at XJH and 1.346 ppbv yr$^{-1}$ at PD. The variation of the median 8-h $O_3$ concentration (ppbv) from
2006 to 2015 averaged for 31 sites over Shanghai (c), also shows the increasing variability of
1.571 ppbv yr$^{-1}$.



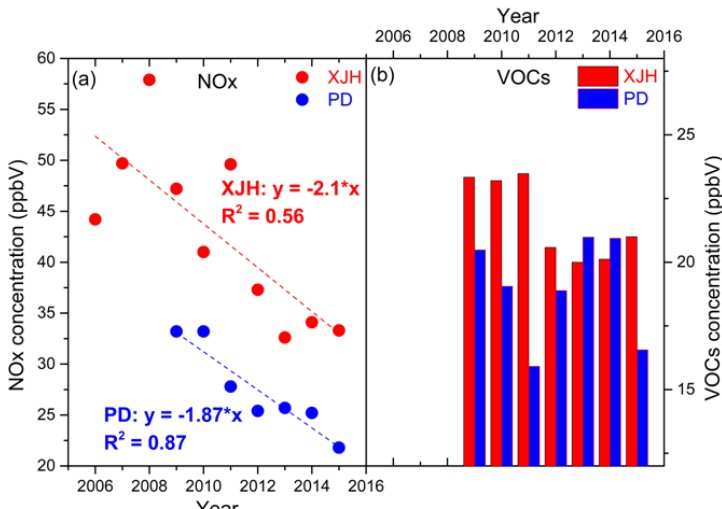


**Figure 3.** The mean annual concentrations (ppbv) of (a) NO$_x$ (dots) and (b) VOCs (bars) from 2006
to 2015 at downtown site XJH and sub-urban site PD respectively. The NO$_x$ concentrations at XJH
and PD both present obvious decreasing trends with 2.1 ppbv yr$^{-1}$ and 1.87 ppbv yr$^{-1}$. While the
VOCs concentrations at both sites present no clear inter-annual trends.



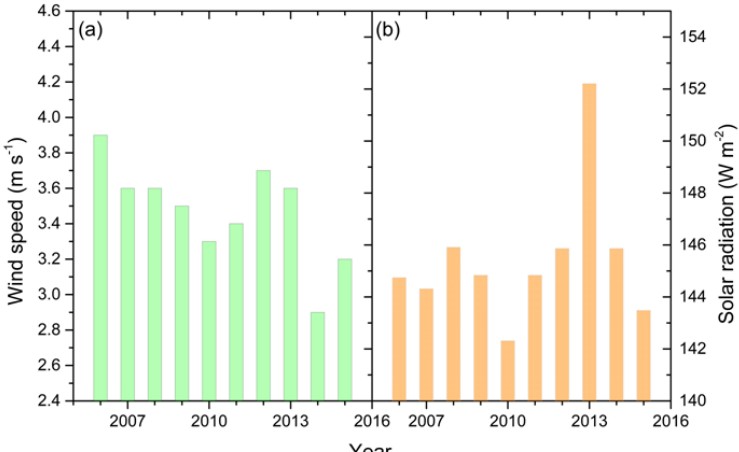


**Figure 4.** The annual variation of (a) summer wind speed (m s$^{-1}$) and (b) total solar radiation (W m$^{-2}$) from 2006 to 2015 in Shanghai. Both wind speed and the solar radiation present weak inter-annual variations but without significant trends.







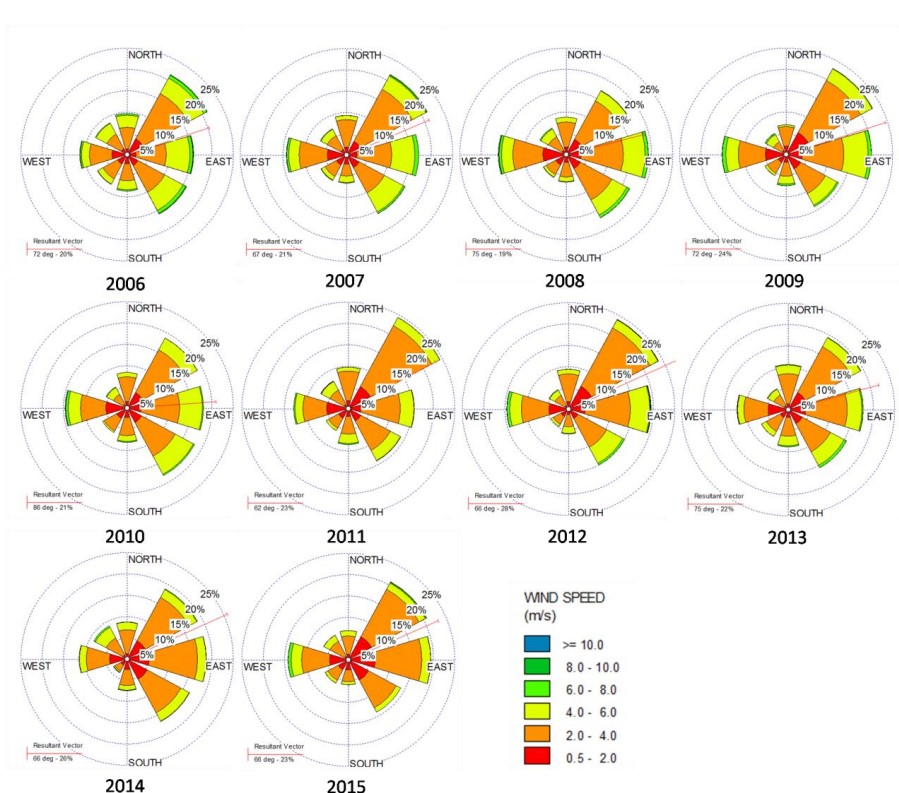


**Figure 5.** The wind rose of each year from 2006 to 2015 in Shanghai. The red line means the
resultant vector suggesting the dominant wind direction.






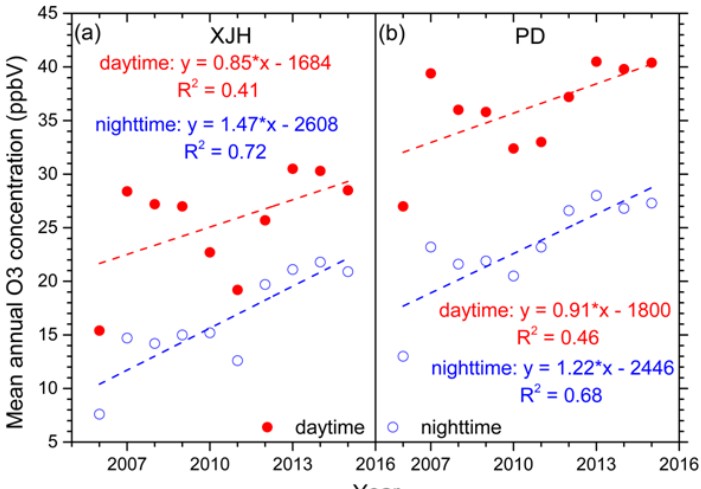


**Figure 6.** The annual variations of daytime and nighttime $O_3$ concentration (ppbv) from 2006 to
2015 at (a) downtown site XJH and (b) sub-urban site PD.


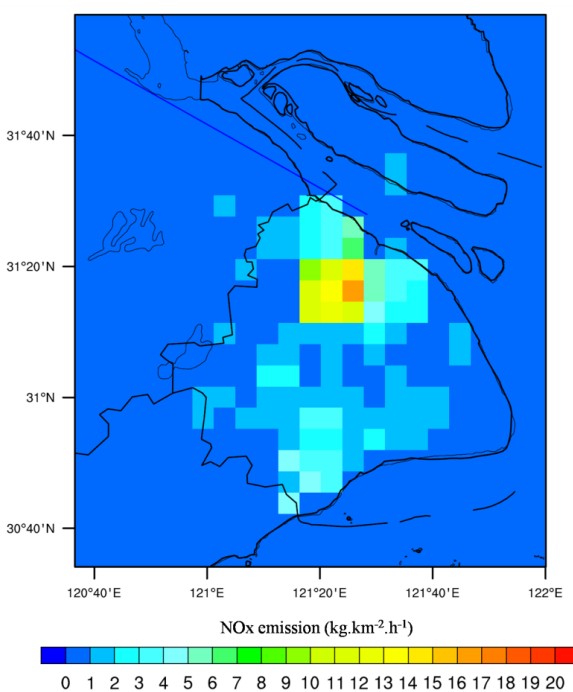


**Figure 7.** The distribution of $NO_x$ emission (kg km$^{-2}$ h$^{-1}$) in 2009 in Shanghai.




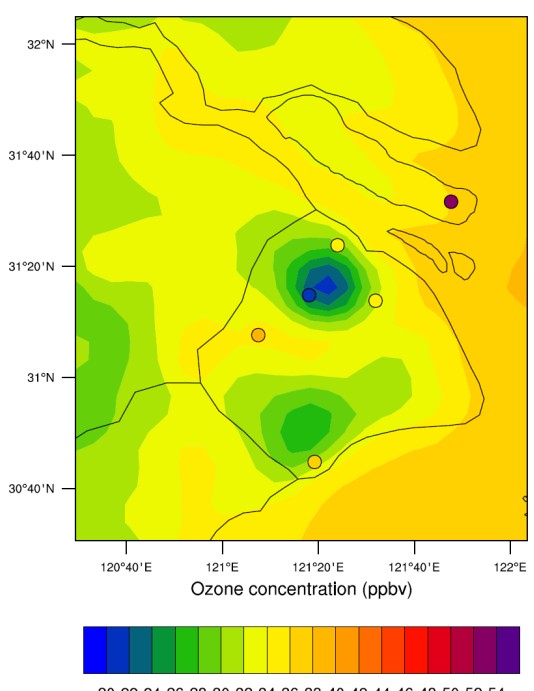

**Figure 8.** The calculated distribution of $O_3$ concentration by WRF-Chem (shade) in September of 2009 compared with measurements (circles) of 6 sites over Shanghai.



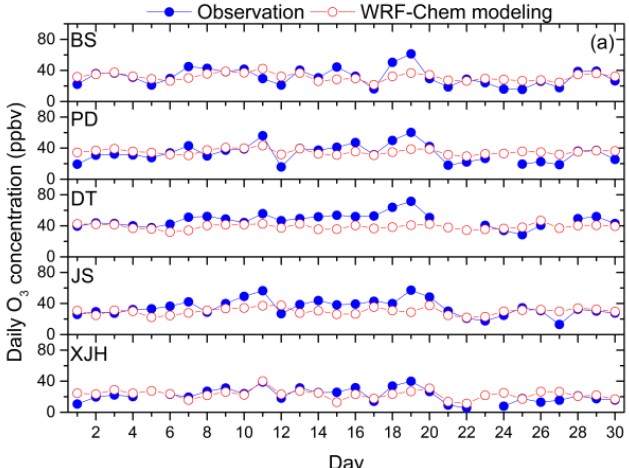


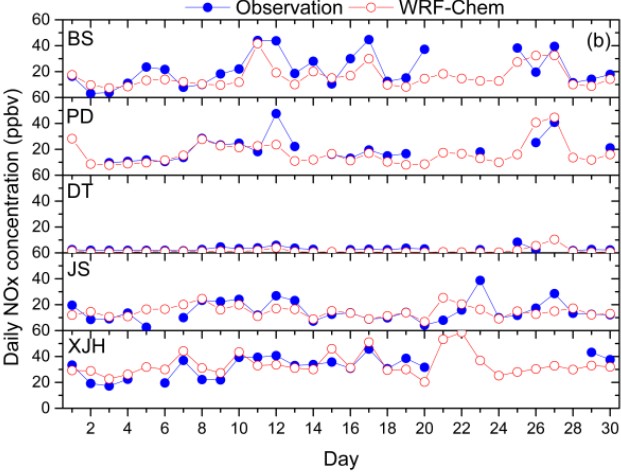


**Figure 9.** The calculated mean daily concentrations (ppbv) of (a) $O_3$ and (b) $NO_x$ at 5 sites in
September of 2009 by WRF-Chem (red circles) and compared with measurements (blue circles).



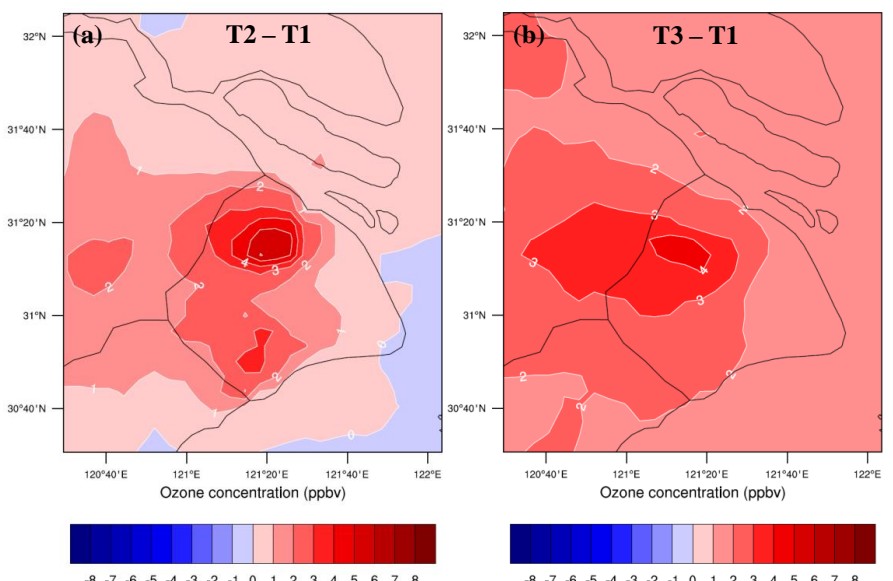

**Figure 10.** The difference of O3 concentration (ppbv) between (a) T2 and T1 (T2-T1), (b) T3 and T1 (T3-T1) respectively conducted by WRF-Chem model. The difference between T2 and T1 lies in the $NO_x$ emissions set in T2 (2015 scenario) is 30% lower than that in T1 (2009 scenario), which is estimated by Lin et al. (2017) according to the Shanghai Environment Yearbook. The difference between T3 and T1 is dependent on that the VOCs emission in T3 is 50% higher than that in T1.

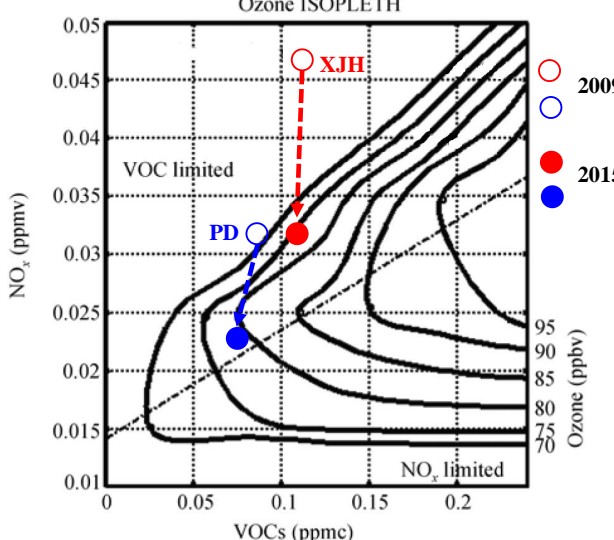

**Figure 11.** The $O_3$ chemical production at downtown site XJH and sub-urban site PD in 2009 and 2015 depicted by $O_3$ isopleths diagram. The hollow and solid red circles denote $O_3$ production regime at XJH in 2005 and 2019 respectively. The hollow and solid blue circles denote $O_3$ production regime at PD in 2005 and 2019 respectively





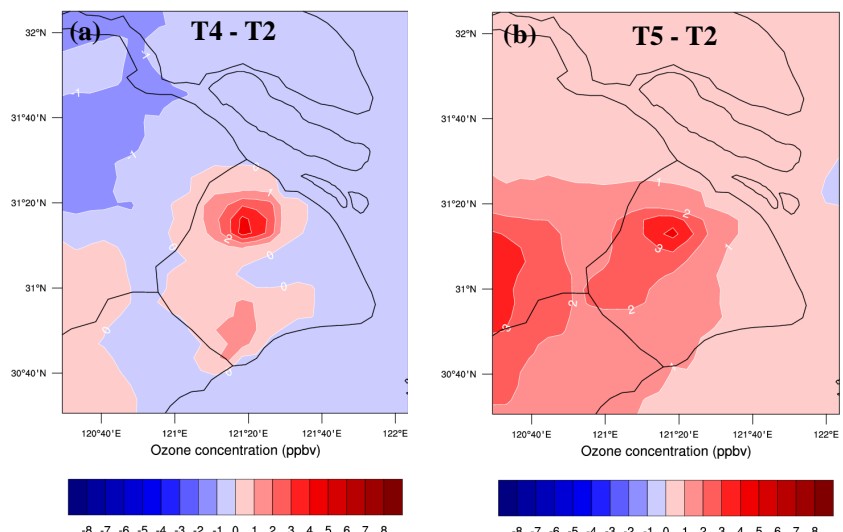

961

**Figure 12.** The difference of O3 concentration (ppbv) between (a) T4 and T2 (T4-T2), (b) T5 and T2 (T5-T2) respectively conducted by WRF-Chem model. The difference between T4 and T2 is that the $NO_x$ emissions set in T4 (2020 scenario) is 20% lower than that in T2 (2015 scenario), which is estimated according to the Shanghai Clean Air Action Plan. The difference between T5 and T2 lies in that the VOCs emission in T5 is 50% higher than that in T2.









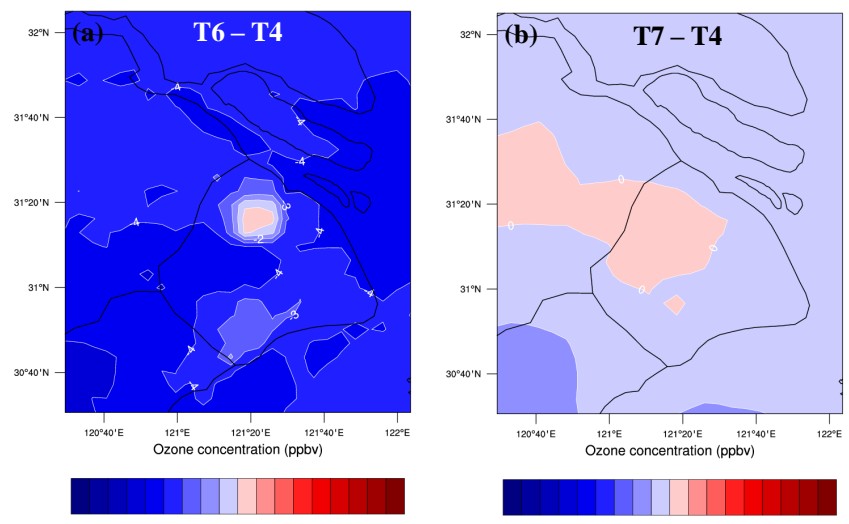

**Figure 13.** The difference of O3 concentration (ppbv) between (a) T6 and T4 (T6-T4), (b) T7 and
T4 (T7-T4) respectively conducted by WRF-Chem model. The $NO_x$ emissions set in T6 is 20% lower
than that in T4 (2020 scenario). The VOCs emission in T7 is 50% higher than that in T4.