# Peer review of "Measurement and model analyses of the ozone variation during 2006 to 2015 and its response to emission change in megacity Shanghai, China Jianming Xu1,2, Xuexi Tie3,4, Wei Gao1,2, Yanfen Lin5, and Qingyan Fu5 1 Yangtze River Delta Cen"

_Atmospheric Chemistry and Physics, 2019_

## Referee Comment (RC1) · Anonymous Referee #1 · 5 Apr 2019

This manuscript studied O3 trend during the past 10 years and ascribed the increasing O3 trend to emission changes (i.e., decrease of NOx emissions) because currently O3 formation in Shanghai is VOC-limited. Following these observational analysis, WRF-Chem simulations for September 2009 with different emission scenarios were conducted to investigate impact of future emission changes on O3 in Shanghai. The sensitivity WRF-Chem simulations indicate that O3 formation in Shanghai may change to NOx-limited at certain point (2020) and O3 may decrease since then if further reduction of NOx emission would happen.

The scientific story is interesting. The structure of the manuscript is logically con-

structed, and the English writing is good. I would recommend publish after addressing the following concerns/comments:

Major comments

1. This manuscript apparently focused on the daytime O3 formation mechanisms (VOC-limited or NOx-limited). But most of the analysis is conducted for daily mean O3 concentration. Daytime O3 and nighttime O3 are affected by totally different processes/mechanisms and they may have different variations/trends.

The tread of daily mean O3 discussed mostly in this manuscript may be dominated by the trend of nighttime O3, which are not governed by the daytime O3 formation mechanisms (VOC-limited or NOx-limited).

Given that this manuscript wants to focus on NOx-limited or VOC-limited, I would recommend the authors change their analysis to focus on daily maximum O3.

2. Showing change of mean diurnal variation of O3 would be helpful to identify the different trends of daytime/nighttime O3. In the few places when daytime/nighttime O3 are separately discussed in this manuscript, the time ranges for the "daytime"/"nighttime" are not specified. Thus, it is hard for this reviewer to judge whether the "daytime" O3 is only affected by O3 formation mechanisms or the "daytime" O3 is still affected by the O3 removal processes during nighttime and early morning. Thus, showing the trend of mean diurnal variation is critical.

3. The current writing sounds like nighttime O3 is only affected by NO titration. Actually nighttime O3 is affected by three main processes, i.e., NO titration, dry deposition and vertical mixing (Hu et al., 2013). Both of the latter two processes are related to nighttime turbulence, which are further related to extent of urbanization. Thus the increasing trend of nighttime O3 may reflect reduced NO titration, as well as enhanced nighttime vertical mixing, or say less stable nighttime boundary layer, which may be induced by enhanced urban effects through the years (Hu et al., 2016).

Other minor comments:

LN33 in -> for

LN48-49, I believe these should be in one sentence.

LN76, what is the definition of "non-attainment days" in terms of O3 in China?

LN81, what is the "Chinese National Ambient Air Quality Stand" in terms of O3?

LN95, anti-correlation for what time period? Nighttime anti-correlation does not indicate VOC-limited mechanism.

LN104 "However, such O3 variation responding to emission change has not been clearly investigated". You just wrote "Gao et al. (2017) reported that O3 concentration in Shanghai downtown increased 67% from 2006 to 2015, whereas NOx concentration decreased about 38%"

LN120, I thought Gao et al. analyzed 10 yr data as in this study.

LN124, high-resolution? Figure 7 looks like very coarse resolution. What is the resolution in Fig. 7?

Fig. 7 should be combined into Fig. 1

LN141-142, why these point sources do not show up on Fig. 7? Please mark these major point sources in Fig. 7

LN153, either remove this sentence or move it to the beginning of this paragraph.

LN193, what aerosol module?

LN197, what is the "anon-traditional SOA module"

LN204, Fig. 7 appears to have a resolution coarser than 6km.

LN329-342, these basical O3 reactions should be put in the introduction, rather than in the results.

LN355-356, or more intensified urbanization in XJH (thus more enhanced downward mixing of O3 (Hu et al., 2013))

References:

Hu, X.-M., Klein, P. M., & Xue, M. (2013). Evaluation of the updated YSU planetary boundary layer scheme within WRF for wind resource and air quality assessments. Journal of Geophysical Research-Atmospheres, 118(18), 10490-10505. 10.1002/jgrd.50823

Hu, X. M., Xue, M., Klein, P. M., Illston, B. G., & Chen, S. (2016). Analysis of Urban Effects in Oklahoma City using a Dense Surface Observing Network. Journal of Applied Meteorology and Climatology, 55(3), 723-741. 10.1175/jamc-d-15-0206.1

---

## Referee Comment (RC2) · Anonymous Referee #2 · 12 Apr 2019

Ozone pollution is an important issue in atmospheric environment study. Focusing this scientific issue, this manuscript presented an interesting finding on the $O_3$ pollution in Shanghai, a megacity in China and its response to $O_3$ precursor emission change based on the 2005-2016 measurements and modeling experiments of September, 2019, which could improve our understanding a very complex process $O_3$ pollution. This manuscript falls within the scope of ACP. I'd like to suggest the minor revisions before it is published as follows:

1) The manuscript analyzed the ozone concentration variation based on the measurement of 2006-2015 in Shanghai with the simulation in September 2009. Please clarify the connection between the measurement and model analyses of the ozone variation and the limitations in conclusions from modeling study.

2) It is better to add the discussions in the measurement analyse on the interannual variations in seasonal cycle (monthly change) of daytime/ nighttime $O_3$ concentrations over 2005-2016.

3) Lines 201-212: Please present the resolution of emission used in WRF-Chem modeling.

4) Please clarify which year the meteorology is used in the modeling experiment of 2020 ozone.

---

## Author Comment (AC1) · 28 May 2019

Responses to Reviewers:

Reviewer 1:

We thank the reviewer for the careful reading of the manuscript and helpful comments. We have revised the manuscript following the suggestions as is described below.

(1) This manuscript apparently focused on the daytime O3 formation mechanisms (VOC-limited or NOx-limited). But most of the analysis is conducted for daily mean O3 concentration. Daytime O3 and nighttime O3 are affected by totally different pro-

[Figure]

cesses/mechanisms and they may have different variations/trends. The tread of daily mean O3 discussed mostly in this manuscript may be dominated by the trend of night-time O3, which are not governed by the daytime O3 formation mechanisms (VOC-limited or NOx-limited). Given that this manuscript wants to focus on NOx-limited or VOC-limited, I would recommend the authors change their analysis to focus on daily maximum O3.

According to the excellent suggestion of the reviewer, we analyzed the variation of daily maximum O3 concentration instead of daily mean O3 concentration in Sect. 3.1. The new results were re-plotted in Fig. 2. It was showed that the annual variation of daily maximum O3 concentration also presented significant increasing trend in XJH and PD sites which was same as that of daily mean O3 concentration. The increasing rate was 0.808 ppbv.yr-1 at XJH site, which was lower than that at PD site of 1.374 ppbv.yr-1. The above new results were included in the revised version. Furthermore, we calculated the variability of daily maximum 8h-O3 concentration, which also exhibited the same increasing trends at the rate of 1.063 and 1.403 ppbv.yr-1 at XJH and PD sites respectively.

Fig. 2, Annual variation of daily maximum O3 concentration during 2006 to 2015 at (a) XJH and (b) PD respectively.

(2) Showing change of mean diurnal variation of O3 would be helpful to identify the different trends of daytime/nighttime O3. In the few places when daytime/nighttime O3 are separately discussed in this manuscript, the time ranges for the "daytime"/"nighttime" are not specified. Thus, it is hard for this reviewer to judge whether the "daytime" O3 is only affected by O3 formation mechanisms or the "daytime" O3 is still affected by the O3 removal processes during nighttime and early morning. Thus, showing the trend of mean diurnal variation is critical.

Thanks for the reviewer's suggestion. We calculate the variability of hourly O3 concentration during 2006 to 2015 to investigate the diurnal variations of O3 trend in Sect.

3.4. The O3 concentration showed increasing trend both in daytime (8:00-18:00, LST) and nighttime (19:00-07:00, LST) at XJH and PD sites. The nighttime O3 increased more significantly than daytime O3 at XJH, with the increasing rate of 1.239 and 0.956 ppbv.yr-1 respectively. While at PD the O3 concentration increased by 1.338 ppbv.yr-1 in daytime which was higher than that in nighttime of 1.028 ppbv.yr-1. In comparison, nighttime O3 presented higher increasing rate at downtown site XJH than that at sub-urban site PD due to more NO emissions at urban center or the enhanced urban effects (Hu et al., 2013). The new results were re-plotted in Fig. 6. In addition, we also compared the seasonal variability of daytime and nighttime O3 concentrations at XJH. The larger O3 variability in nighttime than daytime was observed in spring, summer and autumn. For example, the nighttime O3 concentration increased at 1.341, 1.159 and 1.525 ppbv yr-1 in spring, summer and autumn respectively, which are more significant than that of 1.008, 0.378 and 1.370 ppbv yr-1 in daytime. The variability of winter O3 concentrations in daytime and nighttime are generally close perhaps due to the lower O3 photochemical productions. The seasonal results were plotted in the Fig. 7.

Fig.6 The variability of hourly O3 concentration during 2006 to 2015 at XJH and PD respectively

Fig.7 The variability of daytime (08:00-18:00, LST) and nighttime (19:00-07:00, LST) O3 concentration during 2006 to 2015 at XJH in (a) spring, (b) summer, (c) autumn, (d) winter respectively.

(3) The current writing sounds like nighttime O3 is only affected by NO titration. Actually nighttime O3 is affected by three main processes, i.e., NO titration, dry deposition and vertical mixing (Hu et al., 2013). Both of the latter two processes are related to nighttime turbulence, which are further related to extent of urbanization. Thus the increasing trend of nighttime O3 may reflect reduced NO titration, as well as enhanced nighttime vertical mixing, or say less stable nighttime boundary layer, which may be induced by enhanced urban effects through the years (Hu et al., 2016).
Thank for this comment. We agreed that both dry deposition and nighttime turbulence influenced the nighttime O3 concentration according to the work by Hu et al. (2013). We checked the vertical temperature gradient between 1000 hpa and 950 hpa at 20:00 (LST) in Shanghai to indicate the nighttime turbulence intensity based on sounding data, while presented no significant trend during 2010 to 2015. Furthermore, the PBL height at 20:00 (LST) retrieved from MPL measurements also varied insignificantly (slight decreasing trend) during the same period. Based on the above measurements, the variation of turbulence at night may have only minor contribution to the nighttime O3 increasing in Shanghai. However the effect of dry deposition could not be excluded by lacking of measurements, which need further investigation. Such discussion has been included in Sect. 3.4.

Fig. The retrieved PBL height at 20:00 (LST) from MPL measurements, which presented slight decreasing trend.

Minor Comments (1) LN33, in->for LN48-49, I believe these should be in one sentence. LN153, either remove this sentence or move it to the beginning of this paragraph. LN329-342, these basic O3 reactions should be put in the introduction, rather than in the results. LN355-356, or more intensified urbanization in XJH (thus more enhanced downward mixing of O3 (Hu, et al., 2013)) Thanks for the corrections, which have been revised.

(2) anti-correlation for what time period? Nighttime anti-correlation does not indicate VOC-limited mechanism. The NOx and O3 measurements were strongly anti-correlated during noontime (10:00-13:00, LST), which has been revised.

(1) LN76, what is the definition of "non-attainment days" in terms of O3 in China? LN81, what is the "Chinese National Ambient Air Quality Stand" in terms of O3? The "non-attainment days" of O3 was defined in the ambient air quality standard (GB3095-2012) by the Ministry of Ecology and Environment of the People's Republic of China, providing the condition of daily maximum O3 concentration exceeding 200 ug/m3, or

daily maximum 8h-O3 concentration exceeding 100 ug/m3.

(2) LN104 "However, such O3 variation responding to emission change has not been clearly investigated". You just wrote "Gao et al. (2017) reported that O3 concentration in Shanghai downtown increased 67% from 2006 to 2015, whereas NOx concentration decreased about 38%"; LN120, I thought Gao et al. analyzed 10 yr data as in this study. Gao et al. (2017) has investigated the O3 variation during 2006 to 2015 in Shanghai, while only limited in the downtown XJH site. In this study, we calculated and compared the variability of mean daily maximum O3 concentration, and mean daily maximum 8h-O3 concentration at downtown site XJH and sub-urban site PD by more comprehensive measurements. In addition, we further illustrated the O3 increasing trend in the larger scale by using the O3 measurements from 31 sites over Shanghai, which were not reported by Gao et al. (2017). Furthermore, this study explored the O3 enhancement response to NOx reduction in Shanghai by WRF-Chem models. The effects of emission changes on long term O3 variability were evaluated by WRF-Chem and compared with measurements. In addition, the shift of O3 photochemical regime relative to the variations of NOx and VOCs concentrations in the past ten years was discussed by O3 isopleths diagram combined with WRF-Chem to provide more insights into the O3 control strategy. Moreover, the future O3 levels and its possible chemical regime in Shanghai were also discussed according to the Shanghai Clean Air Action Plan. This was complemented in the part of Introduction.

(3) LN124, high-resolution? Figure 7 looks like very coarse resolution. What is the resolution in Fig. 7? Fig. 7 should be combined into Fig. 1; LN141-142, why these point sources do not show up on Fig. 7? Please mark these major point sources in Fig. 7 LN204, Fig. 7 appears to have a resolution coarser than 6km. The horizontal resolution set in WRF-Chem was 6km. However the emission inventory used in WRF-Chem was extracted and combined from the MEIC (0.25o) and MIRAGE-Shanghai (0.16o) emissions and equally allocated to model grids with 6 km resolution. Thus the emission in Fig.7 seemed to be a little coarse due to the coarse resolution of the

emission inventory data. According to the suggestion from the reviewer, the Fig.7 was combined into Fig.1, and the major point source around BS site was marked in Fig.1.

Fig.1 (a) The distribution of land-use category in Shanghai. The blue dots denote the locations of 6 sties (XJH, BS, PD, SS, JS, DT), in which XJH site is located at the downtown of Shanghai, with large emission from transportation, PD site is located at the sub-urban area with the mixed emissions of transportation and residential, JS site is located in the south of Shanghai with several large chemical industries, BS site is located in the north of Shanghai with some big steel and power plants, SS site is located at the top of the sole hill (100 m a.g.l) with influence from regional transport, DT site is located at a remote island without anthropogenic activities. (b) The NOx emission of 2009 scenario in Shanghai.

(4) LN193, what aerosol module? LN197, what is the "anon-traditional SOA module" The WRF-Chem model used in this study was not the standard version from the WRF DOWNLOAD website. It was mainly improved by Tie et al. (2007) and Li et al. (2010; 2011). The aerosol module was developed by the US EPA (Binkowski and Roselle, 2003) and used in CMAQ model. The secondary organic aerosol (SOA) formation is simulated using a non-traditional SOA model including the volatility basis-set modeling method in which primary organic components are assumed to be semi-volatile and photochemically reactive and are distributed in logarithmically spaced volatility bins (Li et al., 2011). The partitioning of semi-volatile organic species is calculated using the algorithm suggested by Koo et al. (2003), in which the bulk gas and particle phases are in equilibrium and all condensable organics form a pseudoideal solution (Odum et al., 1996). Nine surrogate species with saturation concentrations from $10^{-2}$ to $10^{6}$ $\mu gm^{-3}$ at room temperature are used for the primary organic aerosol (POA) components following the approach of Shrivastava et al. (2008). The SOA contribution from glyoxal and methylglyoxal is also included (Li et al., 2011). These were added to the revised version.

The complete response file including text and figures were in the supplement.

Please also note the supplement to this comment:
https://www.atmos-chem-phys-discuss.net/acp-2019-160/acp-2019-160-AC1-
supplement.pdf

**Supplement:**

**Responses to Reviewers:**

**Reviewer 1:**

We thank the reviewer for the careful reading of the manuscript and helpful comments. We have revised the manuscript following the suggestions as is described below.

(1) This manuscript apparently focused on the daytime O3 formation mechanisms (VOC-limited or NOx-limited). But most of the analysis is conducted for daily mean O3 concentration. Daytime O3 and nighttime O3 are affected by totally different processes/mechanisms and they may have different variations/trends. The tread of daily mean O3 discussed mostly in this manuscript may be dominated by the trend of nighttime O3, which are not governed by the daytime O3 formation mechanisms (VOC-limited or NOx-limited). **Given that this manuscript wants to focus on NOx-limited or VOC-limited, I would recommend the authors change their analysis to focus on daily maximum O3.**

According to the excellent suggestion of the reviewer, we analyzed the variation of daily maximum $O_3$ concentration instead of daily mean $O_3$ concentration in Sect. 3.1. The new results were re-plotted in Fig. 2. It was showed that the annual variation of daily maximum $O_3$ concentration also presented significant increasing trend in XJH and PD sites which was same as that of daily mean $O_3$ concentration. The increasing rate was 0.808 ppbv.yr$^{-1}$ at XJH site, which was lower than that at PD site of 1.374 ppbv.yr$^{-1}$. The above new results were included in the revised version. Furthermore, we calculated the variability of daily maximum 8h-$O_3$ concentration, which also exhibited the same increasing trends at the rate of 1.063 and 1.403 ppbv.yr$^{-1}$ at XJH and PD sites respectively.

[Figure]

Fig. 2, Annual variation of daily maximum $O_3$ concentration during 2006 to 2015 at (a) XJH and (b) PD respectively.

(2) Showing change of mean diurnal variation of O3 would be helpful to identify the different trends of daytime/nighttime O3. In the few places when daytime/nighttime O3 are separately discussed in this manuscript, the time ranges for the "daytime"/"nighttime" are not specified. Thus, it is hard for this reviewer to judge whether the "daytime" O3 is only affected by O3 formation mechanisms or the "daytime" O3 is still affected by the O3 removal processes during nighttime and early morning. Thus, showing the trend of mean diurnal variation is critical.

Thanks for the reviewer's suggestion. We calculate the variability of hourly $O_3$ concentration during 2006 to 2015 to investigate the diurnal variations of $O_3$ trend in Sect. 3.4. The $O_3$ concentration showed increasing trend both in daytime (8:00-18:00, LST) and nighttime (19:00-07:00, LST) at XJH and PD sites. The nighttime $O_3$ increased more significantly than daytime $O_3$ at XJH, with the increasing rate of 1.239 and 0.956 ppbv.yr$^{-1}$ respectively. While at PD the $O_3$ concentration increased by 1.338 ppbv.yr$^{-1}$ in daytime which was higher than that in nighttime of 1.028 ppbv.yr$^{-1}$. In comparison, nighttime $O_3$ presented higher increasing rate at downtown site XJH than that at sub-urban site PD due to more NO emissions at urban center or the enhanced urban effects (Hu et al., 2013). The new results were re-plotted in Fig. 6. In addition, we also compared the seasonal variability of daytime and nighttime $O_3$ concentrations at XJH. The larger $O_3$ variability in nighttime than daytime was observed in spring, summer and autumn. For example, the nighttime $O_3$ concentration increased at 1.341, 1.159 and 1.525 ppbv yr-1 in spring, summer and autumn respectively, which are more significant than that of 1.008, 0.378 and 1.370 ppbv yr-1 in daytime. The variability of winter $O_3$ concentrations in daytime and nighttime are generally close perhaps due to the lower O3 photochemical productions. The seasonal results were plotted in the Fig. 7.

[Figure]

Fig.6 The variability of hourly O₃ concentration during 2006 to 2015 at XJH and PD respectively

[Figure]

Fig.7 The variability of daytime (08:00-18:00, LST) and nighttime (19:00-07:00, LST) O₃ concentration during 2006 to 2015 at XJH in (a) spring, (b) summer, (c) autumn, (d) winter respectively.

(3) The current writing sounds like nighttime O3 is only affected by NO titration. Actually nighttime O3 is affected by three main processes, i.e., NO titration, dry deposition and vertical mixing (Hu et al., 2013). Both of the latter two processes are related to nighttime turbulence, which are further related to extent of urbanization. Thus the increasing trend of nighttime O3 may reflect reduced NO titration, as well as enhanced nighttime vertical mixing, or say less stable nighttime boundary layer, which may be induced by enhanced urban effects through the years (Hu et al., 2016).

Thank for this comment. We agreed that both dry deposition and nighttime turbulence influenced the nighttime O₃ concentration according to the work by Hu et al. (2013). We checked the vertical temperature gradient between 1000 hpa and 950 hpa at 20:00 (LST) in Shanghai to indicate the nighttime turbulence intensity based on sounding data, while presented no significant trend during 2010 to 2015. Furthermore, the PBL height at 20:00 (LST) retrieved from MPL measurements also varied insignificantly (slight decreasing trend) during the same period. Based on the above measurements, the variation of turbulence at night may have only minor contribution to the nighttime O₃ increasing in Shanghai. However the effect of dry deposition could not be excluded by lacking of measurements, which need further investigation. Such discussion has been included in Sect. 3.4.

[Figure]

Fig. The retrieved PBL height at 20:00 (LST) from MPL measurements, which presented slight decreasing trend.

Minor Comments
(1) LN33, in->for
    LN48-49, I believe these should be in one sentence.
    LN153, either remove this sentence or move it to the beginning of this paragraph.
    LN329-342, these basic O3 reactions should be put in the introduction, rather than in the results.
    LN355-356, or more intensified urbanization in XJH (thus more enhanced downward mixing of O3 (Hu, et al., 2013))
Thanks for the corrections, which have been revised.

(2) anti-correlation for what time period? Nighttime anti-correlation does not indicate VOC-limited mechanism.
The NOx and $O_3$ measurements were strongly anti-correlated during noontime (10:00-13:00, LST), which has been revised.

(1) LN76, what is the definition of "non-attainment days" in terms of O3 in China? LN81, what is the "Chinese National Ambient Air Quality Stand" in terms of O3?
The "non-attainment days" of $O_3$ was defined in the ambient air quality standard (GB3095-2012) by the Ministry of Ecology and Environment of the People's Republic of China, providing the condition of daily maximum $O_3$ concentration exceeding 200 ug/m$^3$, or daily maximum 8h-$O_3$ concentration exceeding 100 ug/m$^3$.

(2) LN104 "However, such O3 variation responding to emission change has not been clearly investigated". You just wrote "Gao et al. (2017) reported that O3

concentration in Shanghai downtown increased 67% from 2006 to 2015, whereas NOx concentration decreased about 38%";

LN120, I thought Gao et al. analyzed 10 yr data as in this study.

Gao et al. (2017) has investigated the $O_3$ variation during 2006 to 2015 in Shanghai, while only limited in the downtown XJH site. In this study, we calculated and compared the variability of mean daily maximum $O_3$ concentration, and mean daily maximum 8h-$O_3$ concentration at downtown site XJH and sub-urban site PD by more comprehensive measurements. In addition, we further illustrated the $O_3$ increasing trend in the larger scale by using the $O_3$ measurements from 31 sites over Shanghai, which were not reported by Gao et al. (2017). Furthermore, this study explored the $O_3$ enhancement response to NOx reduction in Shanghai by WRF-Chem models. The effects of emission changes on long term $O_3$ variability were evaluated by WRF-Chem and compared with measurements. In addition, the shift of $O_3$ photochemical regime relative to the variations of NOx and VOCs concentrations in the past ten years was discussed by $O_3$ isopleths diagram combined with WRF-Chem to provide more insights into the $O_3$ control strategy. Moreover, the future $O_3$ levels and its possible chemical regime in Shanghai were also discussed according to the Shanghai Clean Air Action Plan. This was complemented in the part of Introduction.

(3) LN124, high-resolution? Figure 7 looks like very coarse resolution. What is the resolution in Fig. 7?

Fig. 7 should be combined into Fig. 1;

LN141-142, why these point sources do not show up on Fig. 7? Please mark these major point sources in Fig. 7

LN204, Fig. 7 appears to have a resolution coarser than 6km.

The horizontal resolution set in WRF-Chem was 6km. However the emission inventory used in WRF-Chem was extracted and combined from the MEIC (0.25º) and MIRAGE-Shanghai (0.16º) emissions and equally allocated to model grids with 6 km resolution. Thus the emission in Fig.7 seemed to be a little coarse due to the coarse resolution of the emission inventory data. According to the suggestion from the reviewer, the Fig.7 was combined into Fig.1, and the major point source around BS site was marked in Fig.1.

[Figure]

Fig.1 (a) The distribution of land-use category in Shanghai. The blue dots denote the locations of 6 sties (XJH, BS, PD, SS, JS, DT), in which XJH site is located at the downtown of Shanghai, with large emission from transportation, PD site is located at the sub-urban area with the mixed emissions of transportation and residential, JS site is located in the south of Shanghai with several large chemical industries, BS site is located in the north of Shanghai with some big steel and power plants, SS site is located at the top of the sole hill (100 m a.g.l) with influence from regional transport, DT site is located at a remote island without anthropogenic activities. (b) The NOx emission of 2009 scenario in Shanghai.

(4) LN193, what aerosol module?
    LN197, what is the "anon-traditional SOA module"
The WRF-Chem model used in this study was not the standard version from the WRF DOWNLOAD website. It was mainly improved by Tie et al. (2007) and Li et al. (2010; 2011). The aerosol module was developed by the US EPA (Binkowski and Roselle, 2003) and used in CMAQ model. The secondary organic aerosol (SOA) formation is simulated using a non-traditional SOA model including the volatility basis-set modeling method in which primary organic components are assumed to be semi-volatile and photochemically reactive and are distributed in logarithmically spaced volatility bins (Li et al., 2011). The partitioning of semi-volatile organic species is calculated using the algorithm suggested by Koo et al. (2003), in which the bulk gas and particle phases are in equilibrium and all condensable organics form a pseudoideal solution (Odum et al., 1996). Nine surrogate species with saturation concentrations from $10^{-2}$ to 106 μgm$^{-3}$ at room temperature are used for the primary organic aerosol (POA) components following the approach of Shrivastava et al. (2008). The SOA contribution from

glyoxal and methylglyoxal is also included (Li et al., 2011). These were added to the revised version.

---

## Author Comment (AC2) · 28 May 2019

Responses to Reviewers:

Reviewer 2:

We thank the reviewer for the careful reading of the manuscript and helpful comments. We have revised the manuscript following his/her suggestions as is described below.

(1) The manuscript analyzed the ozone concentration variation based on the measurement of 2006-2015 in Shanghai with the simulation in September 2009. Please clarify the connection between the measurement and model analyses of the ozone variation

[Figure]

and the limitations in conclusions from modeling study. First, we found the notable increasing trend of O3 concentration at XJH and PD sites during 2006 to 2015 based on the long term measurements. By excluding the effects of VOCs and meteorology on the measured O3 enhancement, we speculated that the O3 increasing trend in Shanghai was likely attributed to the reduction of NOx concentration as a result of the strong VOCs-limited regime for O3 production according to the previous studies. Then we used the WRF-Chem model to conduct sensitive experiments to demonstrate the abovementioned speculation. The simulated O3 concentration increased from 2009 to 2015 resulted from 30% reduction of NOx emission estimated by Shanghai Environmental Monitoring Center. The increasing rates of O3 trend at downtown site XJH and sub-urban site PD were estimated by WRF-Chem model at 1.06 ppbv yr-1 and 0.96 ppbv yr-1, which was very close to the observed O3 growth variability. Thus we suggested that the observed increasing trend of O3 concentration during the past ten years in Shanghai was mainly attributed to the reduction of NOx emission under the VOC-limited condition for O3 production. However there were some uncertainties and limitations existed in the study. First, in sensitive experiments the NOx emission was cut evenly for all the grids of model domain, that was to say the inhomogeneity of the NOx reduction was not considered in the sensitive experiments by lacking of the emission inventory with higher resolution. Second, the variation of VOCs emission was not taken into account due to the more uncertainties of the current inventory for VOCs. According to the studies of Geng et al. (2007, 2009), O3 production in Shanghai was very sensitive to some VOC species, especially aromatics. Thus the accurate emission of VOCs need to be developed and included in the future study. Third, the same meteorology was used for all WRF-Chem experiments. However the O3 photolysis, advection, and vertical diffusion were strongly affected by meteorology. For example, O3 concentration in Shanghai was depressed in June due to the Meiyu period with great cloud cover inhibiting the photolysis. The summer O3 concentration was mostly affected by the location and intensity of sub-tropical high which was dominant for the photochemical production. Thus the variation of meteorology would be considered and

evaluated in the future studies for more deep investigation. Above discussion has been complemented in the part of Conclusion.

(2) It is better to add the discussions in the measurement analyse on the interannual variations in seasonal cycle (monthly change) of daytime/ nighttime O3 concentrations over 2005-2016. Thanks for the suggestion. The seasonal variability of daytime and nighttime O3 concentrations at XJH site were presented in the new Fig. 7. Both daytime and night O3 concentrations presented increasing trends in all seasons. In comparison, the larger increasing rates of nighttime O3 concentration were observed in spring, summer and autumn than that of daytime O3 concentrations. For example, the nighttime O3 concentrations increased at 1.341, 1.159 and 1.525 ppbv yr-1 in spring, summer and autumn respectively, which were more significant than that of 1.008, 0.378 and 1.370 ppbv yr-1 in daytime. The variability of winter O3 concentrations in daytime and nighttime were generally close perhaps due to the lower O3 photochemical productions. The above results have been included in the Fig.7.

Fig.7 The variability of daytime and night O3 concentration during 2006 to 2015 at XJH in (a) spring, (b) summer, (c) autumn, (d) winter respectively.

(3) Lines 201-212: Please present the resolution of emission used in WRF-Chem modeling. The emission inventory used in WRF-Chem was extracted and combined from MEIC (Zhang et al., 2009) with 0.25o resolution for the domain out of Shanghai and the MIRAGE-shanghai (Tie et al., 2011) with 0.16o resolution for Shanghai area, which has been introduced in Sect. 2.3.

(4) Please clarify which year the meteorology is used in the modeling experiment of 2020 ozone. The meteorology in September of 2009 was used for the all experiments by WRF-Chem considering that it was very close to the climatological condition in Shanghai.

The complete response including text and figures was upload in supplement.

Please also note the supplement to this comment:
https://www.atmos-chem-phys-discuss.net/acp-2019-160/acp-2019-160-AC2-
supplement.pdf
* * *
[Figure]

**Supplement:**

**Responses to Reviewers:**

**Reviewer 2:**

We thank the reviewer for the careful reading of the manuscript and helpful comments. We have revised the manuscript following his/her suggestions as is described below.

(1) The manuscript analyzed the ozone concentration variation based on the measurement of 2006-2015 in Shanghai with the simulation in September 2009. Please clarify the connection between the measurement and model analyses of the ozone variation and the limitations in conclusions from modeling study.

First, we found the notable increasing trend of $O_3$ concentration at XJH and PD sites during 2006 to 2015 based on the long term measurements. By excluding the effects of VOCs and meteorology on the measured $O_3$ enhancement, we speculated that the $O_3$ increasing trend in Shanghai was likely attributed to the reduction of NOx concentration as a result of the strong VOCs-limited regime for $O_3$ production according to the previous studies. Then we used the WRF-Chem model to conduct sensitive experiments to demonstrate the abovementioned speculation. The simulated $O_3$ concentration increased from 2009 to 2015 resulted from 30% reduction of NOx emission estimated by Shanghai Environmental Monitoring Center. The increasing rates of $O_3$ trend at downtown site XJH and sub-urban site PD were estimated by WRF-Chem model at 1.06 ppbv $yr^{-1}$ and 0.96 ppbv $yr^{-1}$, which was very close to the observed $O_3$ growth variability. Thus we suggested that the observed increasing trend of $O_3$ concentration during the past ten years in Shanghai was mainly attributed to the reduction of NOx emission under the VOC-limited condition for $O_3$ production.

However there were some uncertainties and limitations existed in the study. First, in sensitive experiments the NOx emission was cut evenly for all the grids of model domain, that was to say the inhomogeneity of the NOx reduction was not considered in the sensitive experiments by lacking of the emission inventory with higher resolution. Second, the variation of VOCs emission was not taken into account due to the more uncertainties of the current inventory for VOCs. According to the studies of Geng et al. (2007, 2009), $O_3$ production in Shanghai was very sensitive to some VOC species, especially aromatics. Thus the accurate emission of VOCs need to be developed and included in the future study. Third, the same meteorology was used for all WRF-Chem experiments. However the $O_3$ photolysis, advection, and vertical diffusion were strongly affected by meteorology. For example, $O_3$ concentration in Shanghai was depressed in June due to the Meiyu period with great cloud cover inhibiting the photolysis. The summer $O_3$ concentration was mostly affected by the location and intensity of sub-tropical high which was dominant for the photochemical production. Thus the variation of meteorology would be considered and evaluated in the future

studies for more deep investigation. Above discussion has been complemented in the part of Conclusion.

(2) It is better to add the discussions in the measurement analyse on the interannual variations in seasonal cycle (monthly change) of daytime/ nighttime O3 concentrations over 2005-2016.

Thanks for the suggestion. The seasonal variability of daytime and nighttime $O_3$ concentrations at XJH site were presented in the new Fig. 7. Both daytime and night $O_3$ concentrations presented increasing trends in all seasons. In comparison, the larger increasing rates of nighttime $O_3$ concentration were observed in spring, summer and autumn than that of daytime $O_3$ concentrations. For example, the nighttime $O_3$ concentrations increased at 1.341, 1.159 and 1.525 ppbv $yr^{-1}$ in spring, summer and autumn respectively, which were more significant than that of 1.008, 0.378 and 1.370 ppbv $yr^{-1}$ in daytime. The variability of winter $O_3$ concentrations in daytime and nighttime were generally close perhaps due to the lower $O_3$ photochemical productions. The above results have been included in the Fig.7.

[Figure]

Fig.7 The variability of daytime and night $O_3$ concentration during 2006 to 2015 at XJH in (a) spring, (b) summer, (c) autumn, (d) winter respectively.

(3) Lines 201-212: Please present the resolution of emission used in WRF-Chem modeling.

The emission inventory used in WRF-Chem was extracted and combined from MEIC (Zhang et al., 2009) with 0.25º resolution for the domain out of Shanghai and the MIRAGE-shanghai (Tie et al., 2011) with 0.16º resolution for Shanghai area, which has been introduced in Sect. 2.3.

(4) Please clarify which year the meteorology is used in the modeling experiment of 2020 ozone.

The meteorology in September of 2009 was used for the all experiments by WRF-Chem considering that it was very close to the climatological condition in Shanghai.

---

## Author Response (AR2)

Responses to Reviewer:

We thank the reviewer for the careful reading of the manuscript and helpful comments. We have revised the manuscript following the suggestions.

**(1) I nearly suggested major revision, because caption of Fig. 6 is**

**confusing. I thought Fig. 6 is mean diurnal variation of $O_3$ and**

**wondered how come nighttime O3 is higher than daytime $O_3$ in XJH.**

**Please clarify.**

Thanks for pointing out the typo of the figure caption. We have corrected the caption of Fig.6b. The Y-axis is the annual change rate of the diurnal $O_3$ concentration from 2006 to 2015 instead of $O_3$

concentration. Thus the unit in Fig. 6a is not ppbv, but ppbv.yr$^{-1}$, representing the change rate of mean diurnal $O_3$ concentration from

2006 to 2015.

[Figure]

**Figure 6.** (b) The annual change rate of diurnal $O_3$ concentration ($ppbv.yr^{-1}$) from 2006 to 2015 at downtown site XJH (red bars) and sub-urban site PD (blue bars).

**(2) Could you please actually present mean diurnal variations of $O_3$ in 2006 and 2015 in Fig. 6 ?**

Thanks the suggestion. To address the comments of the reviewer, we add the additional figure to descript the mean diurnal variations of $O_3$ concentration in 2006 and 2015 at XJH and PD site in Fig. 6a. It was showed that the maximum and minimum $O_3$ concentrations occur in the afternoon (14-15 pm) and in the early morning (6-7 am), respectively, at both sites. In addition, the diurnal $O_3$ concentrations at XJH and PD all increase significantly from 2006 to 2015. For example, the peak $O_3$ concentration at XJH increases from 21 ppbv to 37 ppbv, meanwhile the minimum $O_3$ concentration rises from 5 ppbv to 14 ppbv exhibiting higher increasing rate. Similar $O_3$ enhancement is also observed at PD

site during the same period. The description has been included in the revised version.

[Figure]

**Figure 6.** (a) The mean diurnal variation of $O_3$ concentration (ppbv)

compared between 2006 and 2015 in XJH (red dots) and PD (blue dots).

**(3) Please also add spatial distributions of daytime $O_3$ and nighttime $O_3$**

**in Fig. 8, in addition to mean. Actually mean $O_3$ can be removed from**

**Fig. 8**

Thanks for the suggestion. We calculated the daytime and nighttime $O_3$

distribution in September 2009 respectively and compared with measurements in Fig. 8. The mean daytime and nighttime $O_3$

concentrations in September 2009 are calculated by WRF-Chem and compared with measurements over 6 sites in Shanghai presented in Fig. 8a and b respectively. Both modeled and measured $O_3$ concentrations in daytime are higher than that in nighttime. The calculated daytime $O_3$ concentration is about 10-18 ppbv higher than that in nighttime in urban region (XJH and PD), which is consistent with the measured difference of 12-14 ppbv. In addition, both model simulations and in-situ measurements in day and nighttime highlight the lower $O_3$ concentration in urban zones than that in suburb. The simulated $O_3$ concentration in downtown is 28-32 and 12-14 ppbv in daytime and nighttime respectively, significantly lower than that at sub-urban (36-38 and 26-28 ppbv in daytime and nighttime respectively) and rural (40-42 and 36-38 ppbv in daytime and nighttime respectively), which are well consistent with the measurements. Above discussion has been included in the revised version.

[revised manuscript text omitted]